# DEER: A Delay-Resilient Framework for Reinforcement Learning with Variable Delays

## Abstract

Classic reinforcement learning (RL) frequently confronts challenges when handling tasks involving delays. These delays introduce a mismatch between the received observations and the subsequent actions to be executed, evidently deviating from the Markov property. Existing approaches usually tackle this issue with end-to-end solutions using state augmentation, often by augmenting the state space with a predefined maximum dimension to accommodate random delays. However, this black-box approach, characterized by incomprehensible intermediate processes and redundant information in augmented states, can result in instability and even undermine the overall performance. To alleviate the delay challenges in RL, we propose DEER (Delay-resilient Encoder-Enhanced RL), a framework that can effectively enhance the interpretability and address the random delay issues. DEER employs a pretrained encoder to encode delayed states along with their variable-length past action sequences due to different delays. Specifically, we leverage delay-free environment datasets to train the encoder and convert delayed states and their corresponding action sequences into hidden states, which serve as novel delay-free states for further policy training. In a variety of delayed scenarios, the trained encoder can smoothly integrate with standard RL algorithms without extra modifications and enhance the delay-solving capability by simply adapting the input dimension of the original algorithms. We evaluate DEER through extensive experiments on Gym and Mujoco, which confirm that DEER is superior to state-of-the-art RL algorithms in both constant and random delay environments.

## 1 Introduction

Deep reinforcement learning has made substantial development in games (Mnih et al., 2013; Silver et al., 2016) and large language models (Ouyang et al., 2022; Carta et al., 2023), where most works are based on the assumption that action execution and state observation occur instantaneously. However, delays are inevitable in real-world tasks such as robotics (Duan et al., 2016; Hwangbo et al., 2017), remote control (Lampe et al., 2014) and distributed communication (Moon et al., 1999). Prior research (Gu & Niculescu, 2003; Dugard & Verriest, 1998) has revealed the substantial impact of delays on an agent's decision process, which not only leads to performance degradation but also holds the potential to induce instability in dynamic systems, posing severe risks in real-world applications. Notably, in self-driving scenarios, even minor delays in the observation and execution modules can markedly amplify the risk of accidents.

Despite the ubiquity of delay as a practical challenge, related research in the domain of RL remains scarce. Existing methods largely fall into two categories: model-free and model-based approaches. Most model-free approaches (Katsikopoulos & Engelbrecht, 2003a; Nath et al., 2021a; Ramstedt & Pal, 2019; Xiao et al., 2020; Schuitema et al., 2010; Agarwal & Aggarwal, 2021; Bouteiller et al., 2021) rely on state augmentation to transform delayed MDPs into equivalent undelayed ones. Though being successful to some extent, their effectiveness is limited by the augmented state space's dimension. On the one hand, fixed input dimension methods are tailored for environments with constant delays, making them unsuitable for new tasks with different or random delays. On the other hand, the dimension of the augmented state space grows linearly with the length of delay, leading to exponential computational requirements and suboptimal policy learning by the agent. By contrast, model-based methods (Walsh et al., 2007; Hester & Stone, 2013; Chen et al., 2021; Firoiu et al., 2018; Derman et al., 2021) aim to predict the current state using the agent's recently

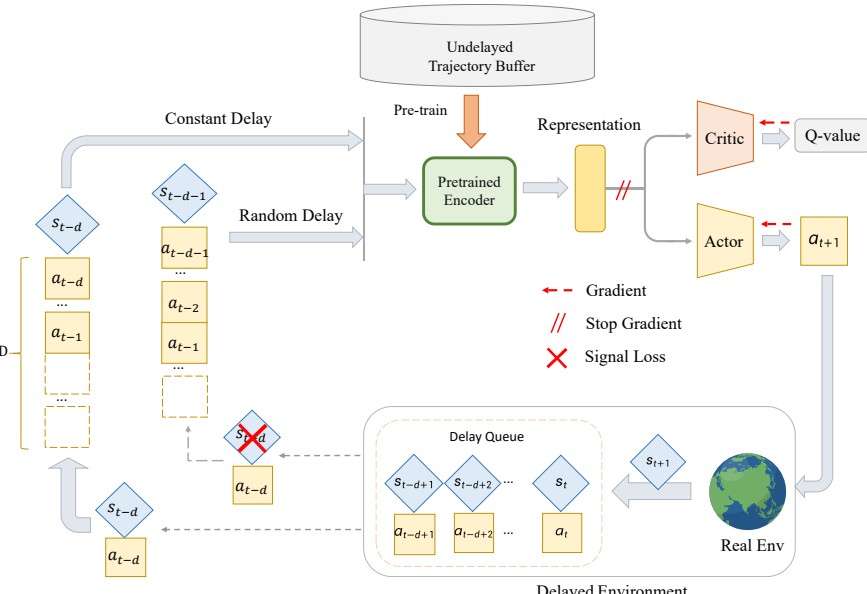

Figure 1: Overview of DEER. The overall process consists of two main parts: pre-trainining an encoder using the offline dataset from undelayed environments to obtain a fixed-length feature representation for the augemnted states, and during the agent's interaction with the delayed environments, utilizing the context representations to guide decision-making. To be specific, details about the process of policy learning are as follows. For environments with a constant delay of $d$, the information state construction is shown by the thick solid line. For random environments, when the agent misses state $s_{t-d}$, the thin dashed line signifies the information state comprising the preceding state and action sequences. The variable D in the figure denotes the maximum delay the agent can tolerate ($D = d_I + d_M$).

received delayed state and action sequence. While being effective in static contexts, their robustness in dynamic environments requires further enhancement. For instance, Firoiu et al. (2018) proposed a predictive model using unrolled Gated Recurrent Unit (GRU) (Chung et al., 2014) modules to iteratively generate a single action, and Derman et al. (2021) introduced the Delayed-Q algorithm for making decisions based on iterative forward dynamic predictions. However, both methods suffer from issues including inference time, model precision, and cumulative errors, that can significantly impact their overall performance.

Considering the outlined challenges, we propose **Delay-resilient Encoder-Enhanced RL (DEER)** that leverages an encoder pretrained on offline datasets to enhance online learning in delayed environments. Instead of making direct decisions using augmented states, we initially map these states into a hidden space known as the context representation space. The actions are subsequently inferred based on these context representations. The overview of DEER is shown in Fig.1. Specifically, we employ an undelayed offline dataset mainly consisting of random trajectories, complemented by a small number of expert trajectories, for the pretraining of an encoder-decoder model. The model's encoder module is designed to generate a context representation that presents a semantic embedding of the delayed state and its corresponding action sequence. This embedding encapsulates the implicit information about both the current state and historical states, effectively serving as a high-dimensional state representation without delay, and can be directly used by standard RL algorithms to generate the current action. This process features three key advantages: (1) The trained encoder can easily generalize to diverse delay environments, as it has been trained across various delay settings. Even when facing an unknown delay in a new environment, the pretrained encoder combined with standard RL algorithms can still work effectively; (2) The proposed approach is versatile in addressing both constant and random delay environments. Since the encoder transforms the augmented state into a fixed-length vector, there is no need to modify the agent's structure for different delay scenarios; (3) This method explicitly breaks down the end-to-end decision process into two distinct stages: encoding the augmented state and making decisions based on the embedding, which significantly improves the interpretability of the entire process. Furthermore, DEER can be seamlessly integrated with any standard RL algorithm. In this paper, we employ Soft Actor-Critic (SAC) as the

decision module, and comprehensive experiments on Gym and Mujoco confirm that our approach is superior to state-of-the-art methods in both constant and random delay environments.

The main contributions of this paper are summarized as follows:

- DEER innovatively leverages offline datasets from delay-free environment tasks to aid in handling tasks occurring within delayed environments.

- A versatile framework DEER is introduced to enhance agent performance in delayed environments, which can be smoothly integrated with standard RL algorithms without any additional modifications.

- With SAC as the decision module, extensive experiments on Gym and Mujoco demonstrate that DEER achieves competitive or superior learning efficiency and performance compared with previous state-of-the-art methods.

## 2 RELATED WORKS

**Offline assisted Online RL**

A large number of works have been done to improve an agent's online performance with the aid of offline RL techniques and they can be categorized as follows:

(1) **Combining offline data with online learning.** Several early works made an attempt to initialize a replay buffer by the demonstration data (Vecerik et al., 2017; Hester et al., 2018), while other works (Lee et al., 2022; Mao et al., 2022; Ball et al., 2023; Nair et al., 2018; Hansen et al., 2022) designed new prioritized sampling schemes to improve learning efficiency and control distribution shift in the online learning stage.

(2) **Pretraining in representation or policy**. The former (Yang & Nachum, 2021) adopted standard contrastive learning methods to extract the features from a variety of offline datasets, which can be applied to downstream tasks including online RL, imitation learning and offline policy optimization. The latter (Rajeswaran et al., 2017; Nair et al., 2020; Zhao et al., 2022; Rudner et al., 2021; Uchendu et al., 2022) called offline-to-online RL has been more prevalent in recent years and commonly executes offline RL algorithms followed by online fine-tuning including parameter transferring, policy regularization, etc.

Our method shares the same key concept with the influential work by Yang & Nachum (2021), yet features significant distinctions in data source, loss function, and working principle. Precisely, we develop an encoder-decoder model to map augmented states, composed of the delayed state and subsequent action sequence, into a common hidden space. This model is trained on a dataset primarily containing random data, with a minor portion of expert data from undelayed environments.

**Encoders in RL**

Encoders have gained widespread usage in reinforcement learning for extracting representations as input to the policy. The RL4Rec framework (Chen et al., 2019; Zhao et al., 2018) employs a state encoder to compress users' historical interactions into a dense representation, capturing user preferences for further inference. Liu et al. (2020) evaluated diverse state encoders and claimed that an attention-based variant can produce the optimal recommendation performance. Generally, encoders in RL4Rec are trained in an end-to-end manner with RL algorithms, distinguishing them from our approach. In visual RL, pretrained encoders are employed to efficiently extract visual features and reduce image input dimensions. Studies such as Shah & Kumar (2021) and Parisi et al. (2022) demonstrated that pretrained ResNet representations can achieve performance comparable to state-based inputs with the aid of expert demonstrations. Additionally, Yuan et al. (2022) investigated the efficacy of the image encoder to enable agents to generalize to unseen visual scenarios with a substantial distributional shift in a zero-shot manner. Moreover, Ge et al. (2021) employed a multi-view state encoder to process input states from multiple perspectives, enhancing generalization abilities via adaptive traffic signal control transfer learning. Nonetheless, the exploitation of pretrained models in delay scenarios remains limited in the current literature.

# 3 PRELIMINARY

## 3.1 MARKOV DECISION PROPERTY (MDP)

The sequential decision-making problem is typically formulated as a discounted Markov Decision Process (MDP), denoted by a tuple $(\mathcal{S}, \mathcal{A}, \rho, p, r, \gamma)$. Here, $\mathcal{S}$ and $\mathcal{A}$ are state and action spaces, respectively; $\rho$ is the initial state distribution; $p : \mathcal{S} \times \mathcal{A} \to \mathcal{S}$ is the transition function; $r : \mathcal{S} \times \mathcal{A} \to \mathbb{R}$ gives the reward to any transitions and $\gamma \in [0, 1)$ is a discount factor. During the interaction between the agent and the environment, the agent follows a policy $\pi : \mathcal{S} \to \mathcal{A}$, resulting in a sequence of transitions or an entire trajectory $\tau = (s_t, a_t, r_t)_{t \geq 0}$. The cumulative return is calculated as $R(\tau) = \sum_{t=0} \gamma^t r_t$ and the primary objective in RL is to identify a return-maxmizing policy $\pi^* = argmax_\pi \mathbb{E}[R(\tau)]$.

## 3.2 RANDOM DROPPING DELAYED MARKOV DECISION PROCESS (RDDMDP)

In real-world scenarios, especially in tasks such as remote control and distributed communication, delays resulting from long-distance transmission or heavy data transfers have a significant impact on agent performance, which are denoted by an intrinsic delay parameter $d_I$ in our work. Morever, during the process of information transmission and interaction, states may be dropped due to obstacles or network malfunctions. Consequently, apart from the first state that is always observable, the instances of state dropout in subsequent steps follow a Bernoulli distribution with parameter $\mu$, representing the probability of dropout. Futhermore, the maximum number of extra dropping steps based on $d_I$, labeled as $d_M$, is defined to ensure that the overall delay is within the agent's capacity limit. Therefore, at each time step $t$, the agent is expected to receive a state $s_{t-d_I}$ and a corresponding reward $r_{t-d_I}$. However, each state dropout follows a Bernoulli distribution: $\omega_t \sim Bern(\mu)$, which implies that when $\omega_t$ equals 0, the agent receives complete information including the state and reward, and when $\omega_t$ is 1, it receives nothing.

As a result, the agent works in an environment with inherent random delays, deviating from the concept discussed in Katsikopoulos & Engelbrecht (2003b) and Nath et al. (2021b). A detailed elaboration on the discrepancies is available in Appendix A.1. The Random Dropping Delayed Markov Decision Process (RDDMDP) is proposed as follows:

**Definition 1** *The RDDMDP can be defined as a 9-tuple $(d_I, d_M, \mathcal{I}_z, \mathcal{A}, \rho, p, r, \gamma, \mu)$:*

*(1) Intrinsic delay value: $d_I \in Z^+$, which is caused by long distance transmission or heavy data transfers;*

*(2) Maximum number of extra dropping steps: $d_M \in Z^+$, which is defined to ensure that $d_I + d_M$ remains within the agent's capacity;*

*(3) Information state space: $\mathcal{I}_z = \mathcal{S} \times \mathcal{A}^z$, where $z$ denotes the random delay value with $d_I \leq z \leq d_I + d_M$, $\mathcal{S}$ and $\mathcal{A}$ are the same as the definition in MDP ;*

*(4) Action space: $\mathcal{A} = \mathcal{A}$;*

*(5) Initial information state distribution: $\rho(i_0) = \rho(s_0, a_0, ..., a_{d_f-1}) = \rho(s_0) \prod_{i=0}^{d_I-1} \delta(a_i - c_i)$, where $\rho$ is the initial state distribution in MDP and $\{c_i\}_{i=0}^{d_I-1}$ are actions selected randomly at the initial of trajectories when states are not observed, and $\delta$ is the Dirac delta function;*

*(6) Transition distribution: $p(i_{t+1}|i_t, a_t)$, where $a_t \in \mathcal{A}$ and the information state $i_t \in \mathcal{I}_z$ is described in detail below;*

*(7) Reward function: $r_t = r_{t-z_t}$, where $z_t$ denotes the random delay value at time $t$;*

*(8) Discount factor: $\gamma \in [0, 1)$;*

*(9) Dropping probability: $\mu \in [0, 1)$, and when $\mu = 0$, the RDDMDP is reduced to the constant delayed MDP (CDMDP) and the details are provided in Appendix A.2.*

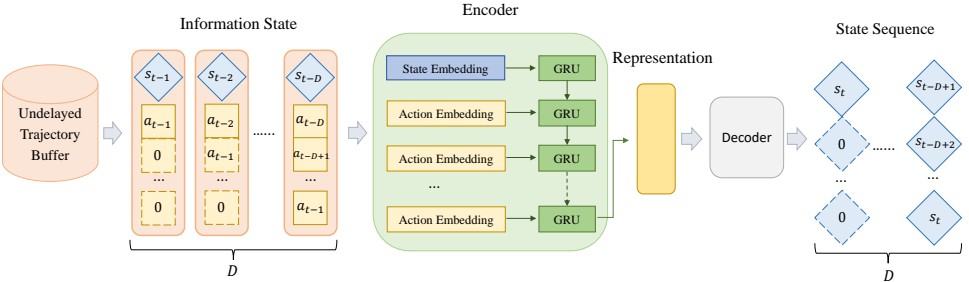

Figure 2: Process of model pretraining. Firstly, the information state dataset is created based on the original undelayed dataset. All state sequences are standardized to a uniform length $D$, where $D$ represents the maximum delay in the environment. Next, these datasets are fed into the seq2seq model and trained in a supervised manner.

At each time $t$, there is a chance of $\mu$ that the agent does not receive the delayed state $s_{t-d_I}$, leading to a potential dropout of state. Thus, the random delay value $z_t$ is defined in the following manner:

$$z_t = \begin{cases} d_I, & \text{if } z_{t-1} = d_I + d_M \text{ or with probability } 1 - \mu, \\ z_{t-1} + 1, & \text{others.} \end{cases}$$

The information state is defined correspondingly as [1]:

$$\begin{aligned} \boldsymbol{i}_t &= (s_{t-z_t}, (a_{t-n}^{(t)})_{n=z_t:1}) \\ &= \begin{cases} (s_{t-d_I}, (a_{t-n}^{(t)})_{n=d_I:1})), & \text{if } z_{t-1} = d_I + d_M \text{ or with probability } 1 - \mu, \\ concatenate(\boldsymbol{i}_{t-1}, a_{t-1}), & \text{others.} \end{cases} \end{aligned}$$

Accordingly, the reward function is expressed as:

$$\begin{aligned} \boldsymbol{r}_t &= r_{t-z_t} \\ &= \begin{cases} r_{t-d_I}, & \text{if } z_{t-1} = d_I + d_M \text{ or with probability } 1 - \mu, \\ \boldsymbol{r}_{t-1}, & \text{others.} \end{cases} \end{aligned}$$

After finishing the aforementioned delay modeling, the agent will continue to take actions based on the current information state $\boldsymbol{i}_t$, akin to its behavior in a delay-free environment.

## 4 METHOD

In this section, we present Delay-resilient Encoder-Enhanced RL (DEER), a concise and effective framework designed to address delays in RL, which capitalizes on the encoder pretrained on undelayed datasets to extract informative features and can properly handle both constant and random delays. The algorithmic framework of DEER is provided in Algorithm 1.

### 4.1 PRETRAINED ENCODER

DEER explicitly utilizes pretrained models as feature extractors, requiring no alteration of the RL algorithm. The pretrained encoder projects information states into embeddings of equal length, helping the agent handle delay challenges without the prior knowledge of environment delays. During the policy learning across training tasks, the encoder's parameters remain fixed to acquire universal information representations.

To acquire a competent encoder, the training of the encoder-decoder model is conducted on the datasets composed of trajectories generated by a random policy along with a few expert trajectories collected by a well-trained SAC agent, all from undelayed environments. The input and output of

---

[1]The superscript of $a_{t_1}^{(t_2)}$ shows that the action is an element of the information state $\boldsymbol{i}_{t_2}$ and the subscript indicates that the action is taken at timestep $t_1$.

the model are referred to as the information state $I_t = (s_t, a_t, ..., a_{t+d-1})$ and the state sequence $(s_{t+1}, ..., s_{t+d})$, respectively, and the encoder-decoder model is employed as a regression model for state predication. In view of the capabilities of the encoder-decoder model, the hidden features extracted by the encoder are expected to contain valuable information about delays, enabling the agent to make proper decisions. Moreover, to make the encoder smoothly generalize across various constant delays and effectively handle random delays, the training dataset consists of information states with diverse action sequence lengths, while maintaining a fixed dimension for the hidden features, so that the encoder can acquire features that can be directly employed by the agent, irrespective of the specific delay conditions.

We use a Seq2Seq (Sutskever et al., 2014) model as the encoder-decoder framework, a simple yet effective choice for handling the delay problem. Multi-Layer Perceptrons (MLPs) are firstly applied to encode each element in the information state, including a state and a series of actions, to generate corresponding embeddings. Subsequently, these embeddings are fed into a GRU module to produce the hidden feature vector whose dimension is a hyperparameter. The Seq2Seq model is optimized based on the MSE loss to improve the accuracy of state sequence predictions, consequently refining the hidden feature's representation of the information state. The complete process of model pretraining is shown in Fig.2, and the detailed network structure and parameter configurations are presented in Appendix B.1.

## 4.2 ENCODER-ENHANCED POLICY LEARNING

The pretrained encoder plays a crucial role in the policy learning phase: extracting essential representations of delayed information and enabling standard RL algorithms to learn effectively regardless of environment delays.

The encoder provides the context representation based on delayed information, offering distinct advantages in both constant and random delay environments. In constant delay settings, its strength lies in the generalization to different types of delays based on the universal training data. This enables direct transformation of information states with unknown lengths into fixed-length representations, avoiding policy input dimension adjustments. In random delay environments, original information states of varying lengths are encoded into hidden features of constant lengths, facilitating the adoption of standard RL algorithms that depend on fixed-length inputs. The entire process of the encoder-enhanced policy learning is shown in Fig.1.

## 5 EXPERIMENTAL RESULTS

In this section, we thoroughly evaluate the effectiveness of our approach by comparing DEER with state-of-the-art RL algorithms in both constant and random delay environments. We investigate various aspects of the context representation's performance within the same scenario, analyzing the impact of decision space dimensions on the final performance. Additionally, we conduct an ablation study to highlight the efficacy of the context representation generated by the pretrained encoder, which is distinct from the predicted state produced by the same model. Moreover, we consider and discuss more factors that influence the experimental outcomes, further elucidating the efficacy of DEER in addressing tasks with delays.

We use SAC (Haarnoja et al., 2018) for decision making, a popular choice for continuous control tasks due to its integration of the actor-critic architecture and the maximum entropy principle. When the context representation is produced by the pretrained encoder, the agent takes the action based on the new state and updates its policy, similar to its behavior in undelayed environments.

All experiments are conducted under the MuJoCo environments from the gym library, including Ant, HalfCheetah, Hopper, Swimmer, Walker2d, and Reacher. Furthermore, each algorithm is executed with 5 different seeds in each environment. The details regarding the number of trajectories used in the pretraining phase are provided in Appendix B.3.

### 5.1 EVALUATION

The following algorithms are used in comparative studies to illustrate the effectiveness of our proposed method:

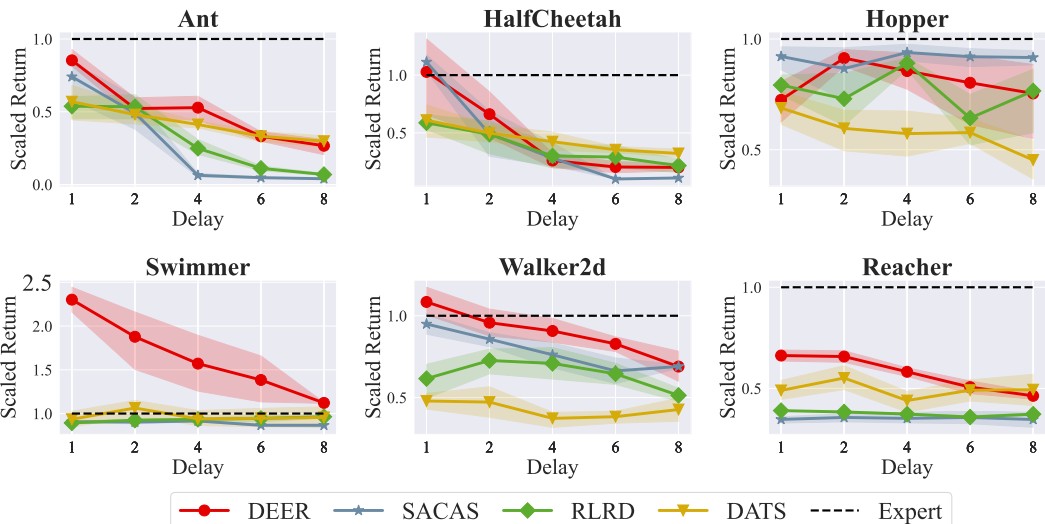

Figure 3: Comparison of algorithms under diverse constant delays.

- **Reinforcement Learning with Random Delays (RLRD; Bouteiller et al., 2021).** RLRD introduces a technique where past actions are relabeled using the current policy. This relabeling procedure generates on-policy sub-trajectories, providing an off-policy and planning-free approach applicable to environments with constant or random delays.

- **Delay-Aware Trajectory Sampling (DATS; Chen et al., 2021).** The effectiveness of DATS can be attributed to the synergistic combination of its unique dynamics model, which incorporates both the known part resulting from delays and the unknown part inherited from the original MDP, and its effective planning method, PETS.

- **Soft Actor-Critic with Augmented States (SACAS).** The implementation of SACAS aligns with the principles described in Katsikopoulos & Engelbrecht (2003a).

Considering the differences in reward settings between DATS and other methods, we normalize the cumulative rewards by $\frac{Return - min\_return}{Expert\_return - min\_return}$. The parameters remain consistent within each algorithm but may vary across different algorithms. $Return$ represents the cumulative rewards obtained in each episode; $min\_return$ corresponds to the minimum return observed throughout all experiments; $Expert\_return$ indicates the level of expertise achieved in undelayed environments.

**Constant Delays.** The initial experiments focus on environments with constant delays. Four algorithms are compared in environments where delay values are set to 1, 2, 4, 6 and 8, respectively. As shown in Figure 3, it is clear that: 1) As delay increases, the performance of all compared algorithms diminishes; 2) In Ant, Swimmer, Walker2d, and Reacher, DEER outperforms other algorithms, evident from their respective performance curves, while in HalfCheetah and Hopper, DEER's performance is similar to that of other algorithms or slightly lower with certain delay values (e.g., Hopper with delay = 8); 3) DEER consistently outperforms the expert in Swimmer across various delays, further highlighting the effectiveness of the context representation in making informed decisions.

**Random Delays.** Randomly delayed environments present a tougher challenge compared with constant delays due to the increased risk of information dropout. We evaluate the four aforementioned algorithms with $d_I = 2$, $d_M = 4$, and dropping probabilities $\mu = 0.2, 0.4$, and $0.6$, respectively. Figures 5 to 7 show the performance comparison of different algorithms under random delays with different $\mu$. To better analyze the results, we take $\mu = 0.2$ and $\mu = 0.4$, and summarize the final results of different algorithms on different tasks in Table 1. Evidently, the loss of information accounts for a notable performance decline across all algorithms, even if they are capable of achieving satisfactory results without random delays. Nevertheless, DEER consistently outperforms its counterparts. In summary, the context representation generated by DEER's pretrained encoder can effectively extract valuable information from delayed states, readily applicable across varying delay settings.

Table 1: Comparison of algorithms with dropping probability 0.2 and 0.4.

| Drop | 0.2 | | | | 0.4 | | | |
|---|---|---|---|---|---|---|---|---|
| Algorithm | DEER | RLRD | DATS | SACAS | DEER | RLRD | DATS | SACAS |
| Ant | 0.47 | 0.16 | 0.23 | 0.08 | 0.38 | 0.02 | 0.17 | 0.03 |
| HalfCheetah | 0.32 | 0.28 | 0.29 | 0.3 | 0.26 | 0.08 | 0.08 | 0.01 |
| Hopper | 0.82 | 0.48 | 0.64 | 0.64 | 0.51 | 0.31 | 0.5 | 0.31 |
| Swimmer | 1.79 | 0.83 | 0.94 | 0.83 | 1.16 | 0.67 | 0.82 | 0.68 |
| Walker2d | 0.62 | 0.49 | 0.44 | 0.26 | 0.32 | 0.34 | 0.35 | 0.158 |
| Reacher | 0.91 | 0.72 | 0.61 | 0.8 | 0.88 | 0.64 | 0.75 | 0.88 |

## 5.2 INFLUENCE OF KEY PARAMETER

The context representation plays a crucial role as the input to the decision model and within the encoder-decoder architecture. Next, we investigate the impact of dimension of the context representation on the agent's performance in delayed environments. The results of DEER for dimensions with 128, 256, and 512 across various delays are presented in Figures 8 - 15. Similarly, to better analyze the impact of context representation dimensions on the performance of the agent under different conditions, we have summarized the results in Tables 2 and 6. From these tables, it can be observed that the performance of DEER in Reacher appears less sensitive to dimension changes and is primarily sensitive to the delay factor. Moreover, in tasks such as HalfCheetah and Swimmer, higher dimensions correspond to improved performance, while Hopper and Walker2d present an opposite trend. This observation suggests that the final performance of the agent depends on the representation capability of the pre-trained encoder, when the training strategy is kept the same. Only when the context representation can well represent the delay information, that is, when the pre-trained encoder can well represent the information state, is it beneficial to the agent's decision-making. Therefore, the dimension of the representation information is not necessarily related to the final performance. In summary, taking into account factors including computational complexity and overall performance, opting for a 256-dimensional context representation is generally recommended.

Table 2: Comparison of DEERs performance in various dimensions with delay values of 4, 6 and 8.

| Delay | 4 | | | 6 | | | 8 | | |
|---|---|---|---|---|---|---|---|---|---|
| Dimension | 128 | 256 | 512 | 128 | 256 | 512 | 128 | 256 | 512 |
| Ant | 1344 | 2574 | 2415 | 889 | 1653 | 1932 | 617 | 1072 | 1381 |
| HalfCheetah | 5236 | 5780 | 5337 | 3320 | 3853 | 4234 | 1874 | 2924 | 3111 |
| Hopper | 2713 | 2918 | 2198 | 2197 | 2565 | 1737 | 1908 | 2462 | 1891 |
| Swimmer | 46 | 78 | 118 | 50 | 83 | 105 | 44 | 48 | 86 |
| Walker2d | 3600 | 4119 | 2098 | 2712 | 3546 | 677 | 1011 | 3074 | 239 |
| Reacher | -7.9 | -7.9 | -8 | -9.9 | -9.8 | -9.9 | -11.7 | -11.5 | -11.5 |

## 5.3 ABLATION STUDY

The ablation study aims to demonstrate the importance of the context representation compared with the predicted state, termed "Decision on Last Predicted State" (DOLPS). DOLPS can be inferred from the decoder module trained with the encoder in DEER at the same time. Experimental results in Figures 16, 17 and Table 3 consistently confirm DEER's advantage over DOLPS across various environments and delays, with DOLPS showing limited effectiveness in Ant, Hopper, and Walker2d. It is clear that the context representation effectively mitigates prediction errors , which is crucial for decisions in the original decision space. Furthermore, it captures historical information embedded in delayed states and action sequences, which is shown to be advantageous for decision-making in the context of delayed scenarios.

Table 3: Comparison of DEER and DOLPS under delay values of 4 and 6.

| Delay | 4 | | 6 | |
|---|---|---|---|---|
| Algorithm | DEER | DOLPS | DEER | DOLPS |
| Ant | 2574 | -10.8 | 1653 | -16 |
| HalfCheetah | 5780 | 2975 | 3853 | 1574 |
| Hopper | 2918 | 663 | 2565 | 652 |
| Swimmer | 78 | 42 | 83 | 45 |
| Walker2d | 4119 | 238 | 3546 | 264 |
| Reacher | -7.9 | -17 | -9.8 | -21 |

## 5.4 MORE ANALYSIS ON DEER

In this section, we'll delve deeper into DEER from six aspects related to its design, execution, and outcomes, aiming to highlight its effectiveness in handling tasks with delays. These aspects encompass various comparisons and discussions: a time performance contrast between DEER and other algorithms, a comparison between online and offline DEER, an analysis against state-of-the-art algorithms in Offline to Online RL, a discussion on the impact of different context representation dimensions on agent performance, a showcase of the effects of three distinct offline datasets on resolving delayed tasks, and a fresh comparison in an alternative scenario of random delays. All of those are provided in Appendix C.4.

## 5.5 LIMITATION

The experimental results confirm DEER's efficacy in addressing delay problems, especially highlighting the significant performance gains achieved by well-pretrained encoders. However, pretrained encoders show certain level of sensitivity to the quantity and state distribution of trajectories used for training. During the process of model pretraining, we carefully selected the number and the type of trajectories based on the specific task to train a better encoder. In-depth discussion and analysis are provided in Appendix C.5.

## 6 CONCLUSION AND FUTURE WORK

In this paper, we introduce DEER, a concise framework designed to effectively tackle delay issues in RL, including both constant delays and random delays, and enhance the interpretability of the entire process. In DEER, an encoder is pretrained using trajectories collected from delay-free environments to map augmented states containing the delayed information into hidden features called context representation, which is subsequently used by the agent to derive new actions. Experiments on DEER combined with SAC demonstrate that our method achieves competitive or superior learning efficiency and performance in comparison with state-of-the-art methods, which validate the effectiveness and efficacy our approach in addressing delay-related challenges.

Future work will focus on extending DEER to visual reinforcement learning, where agents receive and process visual information as states. Additionally, efforts will be made to deploy our approach to real-world systems, such as remote control systems or physical robots, further assessing its performance and applicability in practical scenarios.

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

## A    MORE DISCUSSION

### A.1    DIFFERENCE BETWEEN RDMDP AND RDDMDP

The RDMDP proposed in (Katsikopoulos & Engelbrecht, 2003b) and (Nath et al., 2021b) introduces stochasticity by considering variable time steps between two successive variables, including observations, actions and rewards. The value of these time steps is not fixed and can differ from unity. Additionally, the observation of a later state $s_{t+1}$ can only occur after the observation of the current state $s_t$. This implies that the state assumed to be observed at time $t$ is actually observed at a later time $t + d$, where $d$ represents the accumulated delay values before time $t$ and it gradually increases over time. However, the RDDMDP we study extends the constant delay scenario by introducing a bounded number of steps with missing states. In our settings, if the state $s_{t-d_I}$ was observed at time $t$, the next state and reward would be $s_{t-d_I+1}$ and $r_{t-d_I+1}$ with probability $1 - \mu$ while such information could not be acquired with probability $\mu$ and we would replace them with $s_{t-d_I}$ and $r_{t-d_I}$. Therefore, in this paper, the random delay value is defined as the number of time steps between two observed states. If the next state received in RDDMDP precisely corresponded to the state at the subsequent moment in time, RDDMDP would be equivalent to RDMDP.

### A.2    CONSTANT DELAYED MARKOV DECISION PROCESS (CDMDP)

The dropping probabilty $\mu = 0$ means that there is no information dropout during the interaction between the environment and the agent. Consequently, the RDDMDP can simplified as the Constant Delayed MDP (CDMDP) (Walsh et al., 2009).

**Definition 2** *Similar to RDDMDP $(d_I, d_M, \mathcal{I}_z, \mathcal{A}, \rho, p, r, \gamma, \mu)$, CDMDP can be defined as a 6-tuple $(\mathcal{I}_d, \mathcal{A}, \rho, p, r, \gamma)$:*

*(1) Information state space: $\mathcal{I}_d = \mathcal{S} \times \mathcal{A}^d$, where $d$ denotes the delay step and $i_t = (s_{t-d}, (a_{t-n}^{(t)})_{n=d:1}) \in \mathcal{I}_d$;*

*(2) Action space: $\mathcal{A} = \mathcal{A}$;*

*(3) Initial information state distribution: $\rho(i_0) = \rho(s_0, a_0, ..., a_{d-1}) = \rho(s_0) \prod_{i=0}^{d-1} \delta(a_i - c_i)$, where $(c_i)_{i=0:d-1}$ denotes the initial action sequence and $\delta$ is the Dirac delta function;*

*(4) Transition distribution: $p(i_{t+1}|i_t, a_t) = p(s_{t-d+1}, a_{t-d+1}^{(t+1)}, ..., a_t^{(t+1)}|s_{t-d}, a_{t-d}^{(t)}, ..., a_{t-1}^{(t)}, a_t) = p(s_{t-d+1}|s_{t-d}, a_{t-d}) \prod_{i=1}^{d-1} \delta(a_{t-d+i}^{(t+1)} - a_{t-d+i}^{(t)}) \delta(a_t^{(t+1)} - a_t)$;*

*(5) Reward function: $r_t = r_{t-d}$;*

*(6) Discount factor: $\gamma \in [0, 1)$.*

## B    IMPLEMENTATION DETAILS

### B.1    DETAILS OF THE PRETRAINED MODEL

In this section, we present the detailed settings for the Encoder. The complete structure of the pretrained model is shown in Fig.4. Particularly, in the encoder module, the relevant state representation is computed as follows:

$$h_1 = \text{GRU}_{\text{en}}(\text{MLP}_{S_1}(s_t)),$$
$$h_i = \text{GRU}_{\text{en}}(\text{MLP}_A(a_{t+i-2}), h_{i-1}), \quad i = 2, 3, ..., d + 1,$$

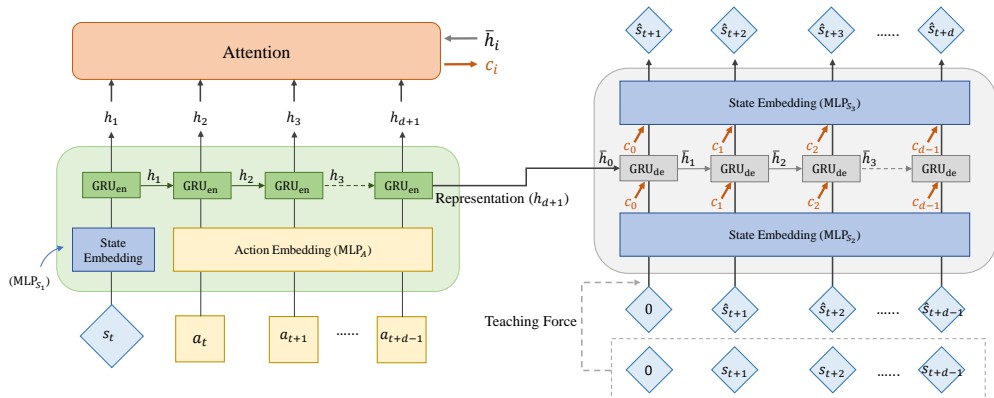

Figure 4: Network structure of pretrained model. It comprises two key modules: the encoder module and the decoder module. In the encoder module, the context representation is generated by first encoding the state and the action sequence within the information state using $\text{MLP}_{S_1}$ and $\text{MLP}_A$, respectively. The resulting encodings are then fed into the $\text{GRU}_{\text{en}}$ to obtain hidden states. The decoder module, on the other hand, is responsible for restoring the encoded information. The state sequence is processed by $\text{MLP}_{S_2}$, followed by inputting it into $\text{GRU}_{\text{de}}$ along with the attention $c_i$ and hidden state $\bar{h}_i$. The final states are achieved by applying $\text{MLP}_{S_3}$. Teaching Force is employed within the decoder to enhance training efficiency.

where $h_{d+1}$ is the representation utilized in the policy training phase and the subscript $d$ denotes the delay value. In the decoder, we use teacher forcing and attention to strengthen the performance,

$$\bar{h}_0 = h_{d+1},$$
$$c_i = \text{Attention}((h_1, ..., h_{d+1}), \bar{h}_i),$$
$$\bar{h}_1 = \text{GRU}_{\text{de}}(\text{MLP}_{S_2}(0) \oplus c_0, \bar{h}_0),$$
$$\bar{h}_i = \begin{cases} \text{GRU}_{\text{de}}(\text{MLP}_{S_2}(\hat{s}_{t+i-1}) \oplus c_{i-1}, \bar{h}_{i-1}), & \text{with probability } p, \\ \text{GRU}_{\text{de}}(\text{MLP}_{S_2}(s_{t+i-1}) \oplus c_{i-1}, \bar{h}_{i-1}), & \text{with probability } 1-p, \end{cases} \quad i = 2, 3, ..., d,$$
$$\hat{s}_{t+i} = \text{MLP}_{S_3}(\bar{h}_i \oplus c_{i-1}), \ i = 1, 2, ..., d,$$

where $p$ denotes the teacher forcing ratio and $\oplus$ denotes the concatenation of tensor along the last dimension.

The parameters of the pretrained model are shown in Table 4. $K_1$ and $K_2$ are both hyperparameters, which denote the dimension of hidden states in GRU and the dimension of embeddings in MLP separately. In this paper, we conduct experiments with $K_1$=128, 256, 512, respectively with $K_2$ = 64.

Table 4: Parameters of the pretrained model

| Name | | Parameters |
|------|---------|------------|
| $\text{GRU}_{\text{en}}$ | 1 Layer | $(K_2, K_1)$ |
| $\text{GRU}_{\text{de}}$ | 1 Layer | $(K_1 + K_2, K_1)$ |
| $\text{MLP}_{S_1}$ | 1 Layer | (state dimension, $K_2$) |
| $\text{MLP}_A$ | 1 Layer | (action dimension, $K_2$) |
| $\text{MLP}_{S_2}$ | 1 Layer | (state dimension, $K_2$) |
| $\text{MLP}_{S_3}$ | 1 Layer | ($2K_1$, state dimension) |

## B.2 PSEUDOCODE

---

**Algorithm 1** Delay-resilient Encoder-Enhanced Reinforcement Learning(DEER)

---

**Input:** $M$ random trajectories, $N$ expert trajectories($M >> N$), maximum number of delayed steps $D$.
**Output:** Policy $\pi_\theta$ in delayed environment.
 1: **Stage 1(Pretraining Model):**
 2: Convert $M + N$ trajectories into training and test datasets, following the data format: $(s_{t-D}, a_{t-D}, \cdots, a_{t-1})$ and corresponding labels: $(s_{t-D+1}, ..., s_t)$, where datasets contain various delays from 1 to $D$ and action sequence in the information state is padded with zeros until its length matches $D$.
 3: Initialize a Seq2Seq Model comprising $Encoder(\cdot)$ and $Decoder(\cdot)$.
 4: Train the Seq2Seq Model using supervised learning on the constructed dataset.
 5: Output the $Encoder(\cdot)$ from the trained Seq2Seq Model.
 6:
 7: **Stage 2 (Decision Model):**
 8: **for** each iteration **do**
 9:     **for** each environment step **do**
10:         Identify current delayed step $d$.
11:         Obtain the information state $I_t = (s_{t-d}, a_{t-d}, ..., a_{t-1})$.
12:         Compute context_representation $h_t = Encoder(I_t)$.
13:         Take action $a_t = \pi_\theta(h_t)$ in the environment, obtain next state $s_{t-d+1}$, and reward $\boldsymbol{r}_t$.
14:         Calculate context_representation $h_{t+1} = Encoder(I_{t+1})$.
15:         Store the transition $(h_t, a_t, \boldsymbol{r}_t, h_{t+1})$ in the replay buffer $\boldsymbol{R}$.
16:         **if** $len(\boldsymbol{R}) \geq training\_threshold$ **then**
17:             Update policy $\pi_\theta$ using the Soft Actor-Critic (SAC) method.
18:         **end if**
19:     **end for**
20: **end for**
21: Output policy $\pi_\theta$ in delayed environments.

---

## B.3 DATASETS USED DURING THE PRETRAINING PROCESS

Table 5: Datasets to pretrain the Seq2Seq model

| Environment | Random Trajs | Expert Trajs |
|---|---|---|
| Ant | 100 | 10 |
| HalfCheetah | 500 | 10 |
| Hopper | 800 | 10 |
| Swimmer | 500 | 10 |
| Walker2d | 8000 | 60 |
| Reacher | 2000 | 10 |

# C ADDITIONAL EXPERIMENTS

## C.1 RANDOM DELAYS

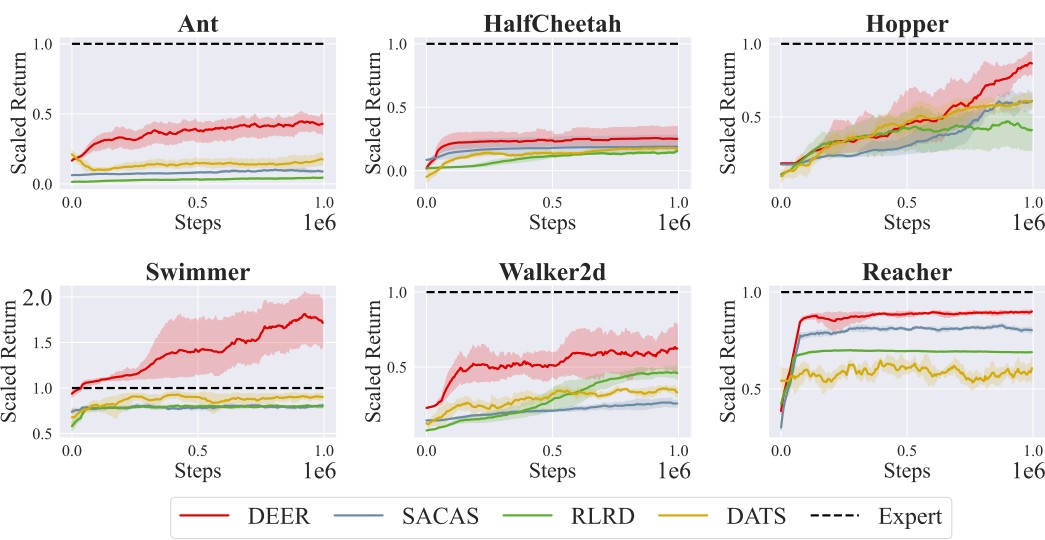

Figure 5: Comparison of algorithms with dropping probability 0.2.

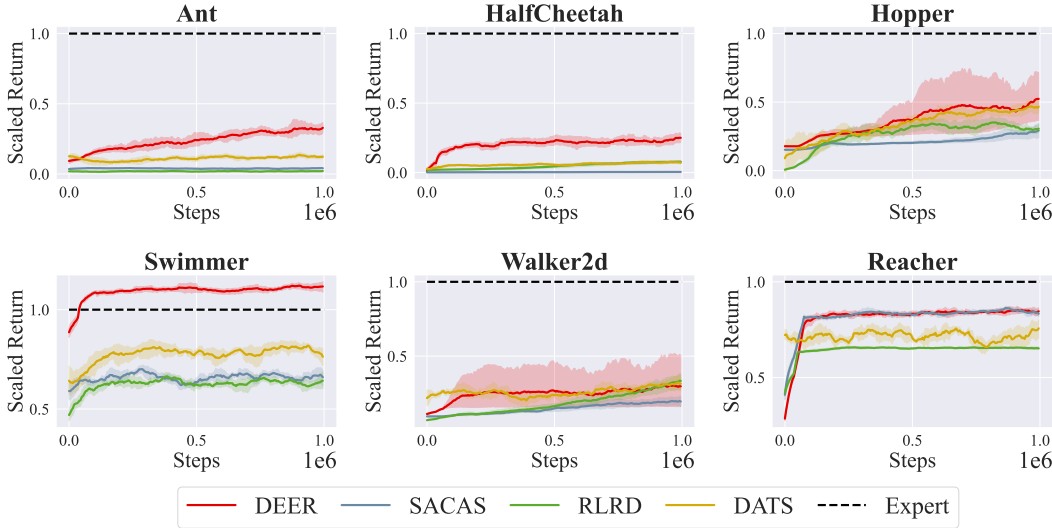

Figure 6: Comparison of algorithms with dropping probability 0.4.

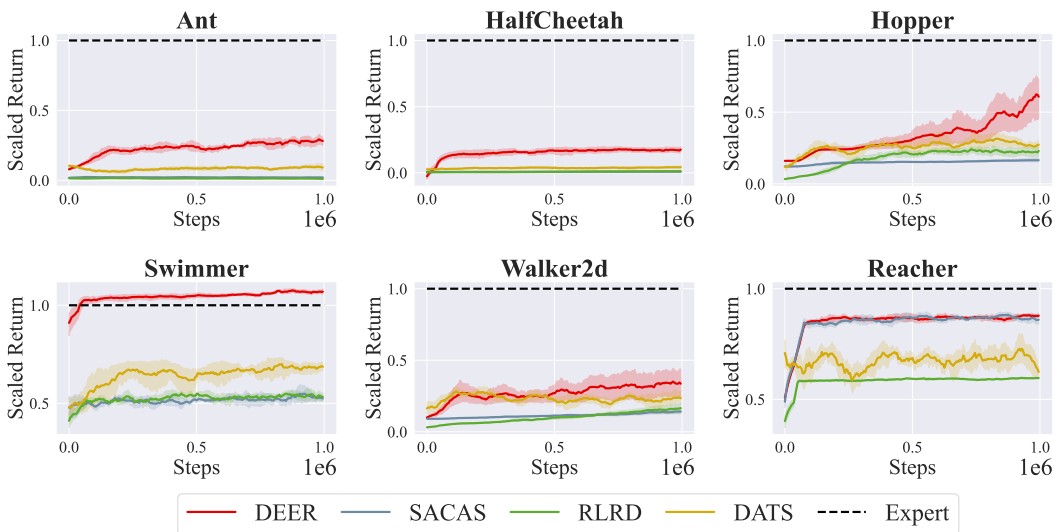

Figure 7: Comparison of algorithms with dropping probability 0.6.

## C.2 KEY PARAMETERS

Table 6: Comparison of DEERs performance in various dimensions with dropping probabilities of 0.2, 0.4 and 0.6.

| Drop | 0.2 | | | 0.4 | | | 0.6 | | |
|---|---|---|---|---|---|---|---|---|---|
| Dimension | 128 | 256 | 512 | 128 | 256 | 512 | 128 | 256 | 512 |
| Ant | 1620 | 2500 | 1632 | 1148 | 1549 | 340 | 711 | 1286 | -12 |
| HalfCheetah | 6362 | 5775 | 6308 | 5036 | 5200 | 5283 | 3692 | 4245 | 5019 |
| Hopper | 2362 | 2661 | 2209 | 2623 | 2337 | 2537 | 2420 | 1961 | 2413 |
| Swimmer | 49 | 90 | 98 | 44 | 47 | 72 | 45 | 46 | 47 |
| Walker2d | 4609 | 4818 | 3556 | 2493 | 4274 | 3467 | 1709 | 2835 | 1394 |
| Reacher | -7 | -6.9 | -7 | -7.8 | -7.8 | -7.5 | -8.6 | -8.5 | -8.5 |

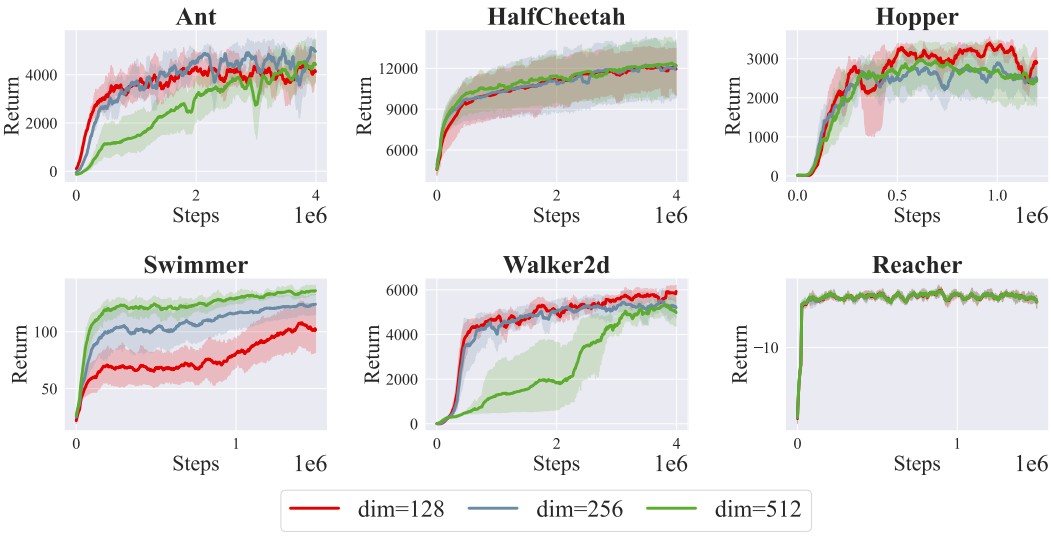

Figure 8: Comparison of DEER's performance with various dimensions and a delay value of 1.

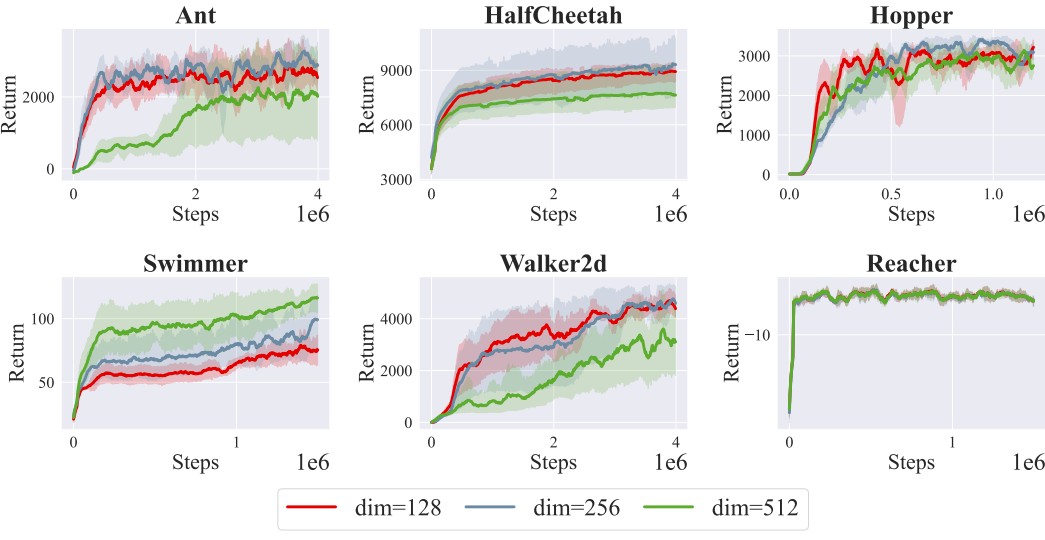

Figure 9: Comparison of DEER's performance with various dimensions and a delay value of 2.

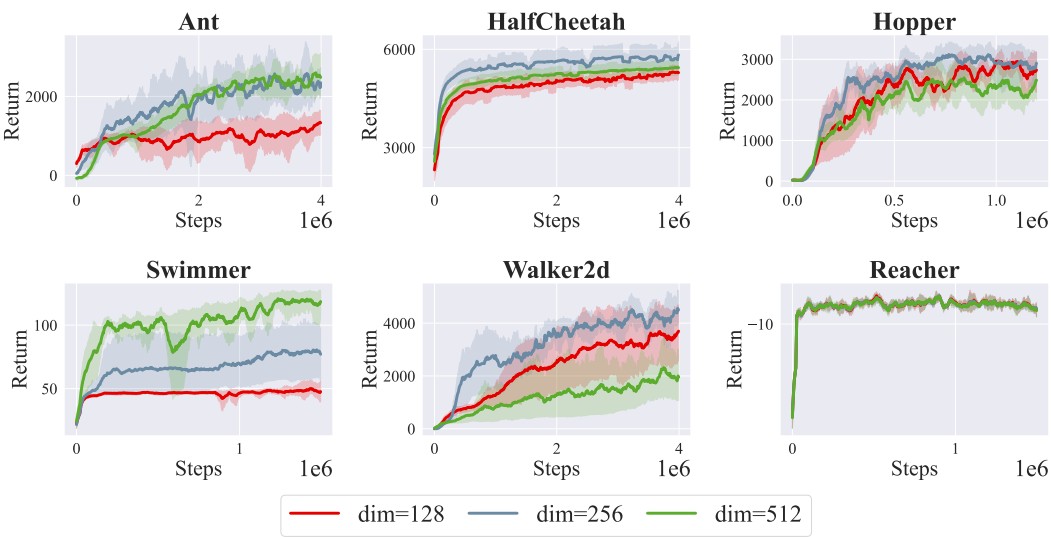

Figure 10: Comparison of DEER's performance with various dimensions and a delay value of 4.

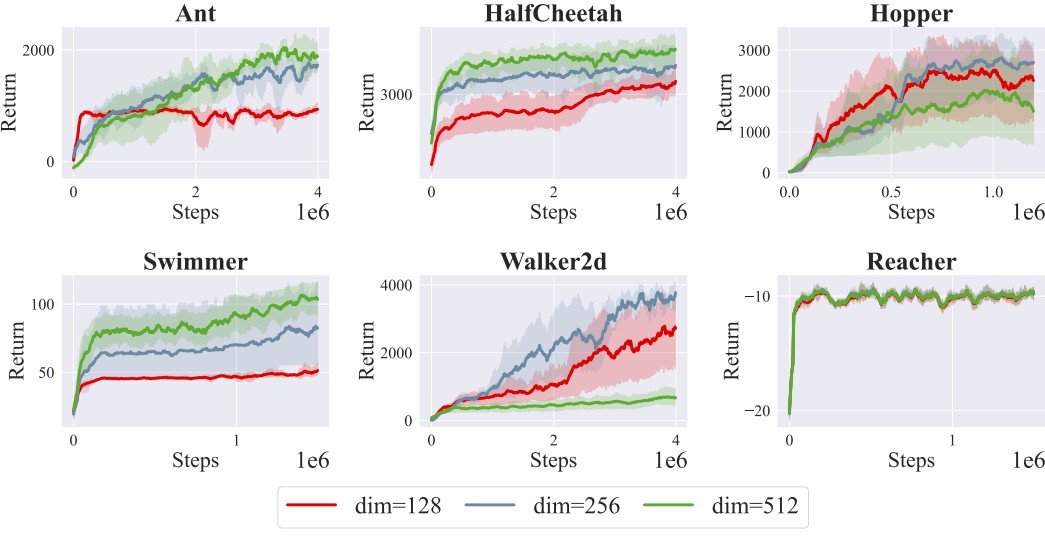

Figure 11: Comparison of DEER's performance with various dimensions and a delay value of 6.

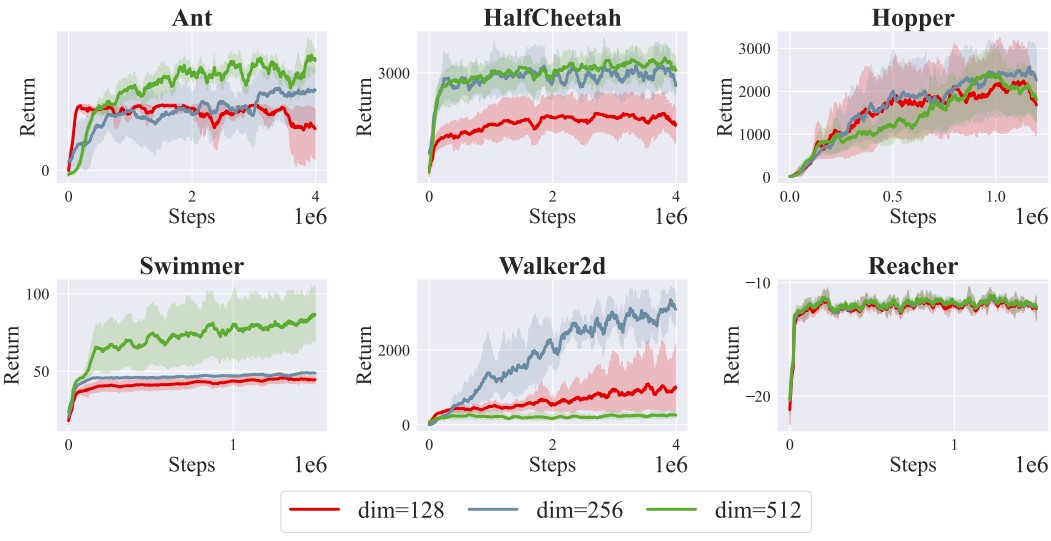

Figure 12: Comparison of DEER's performance with various dimensions and a delay value of 8.

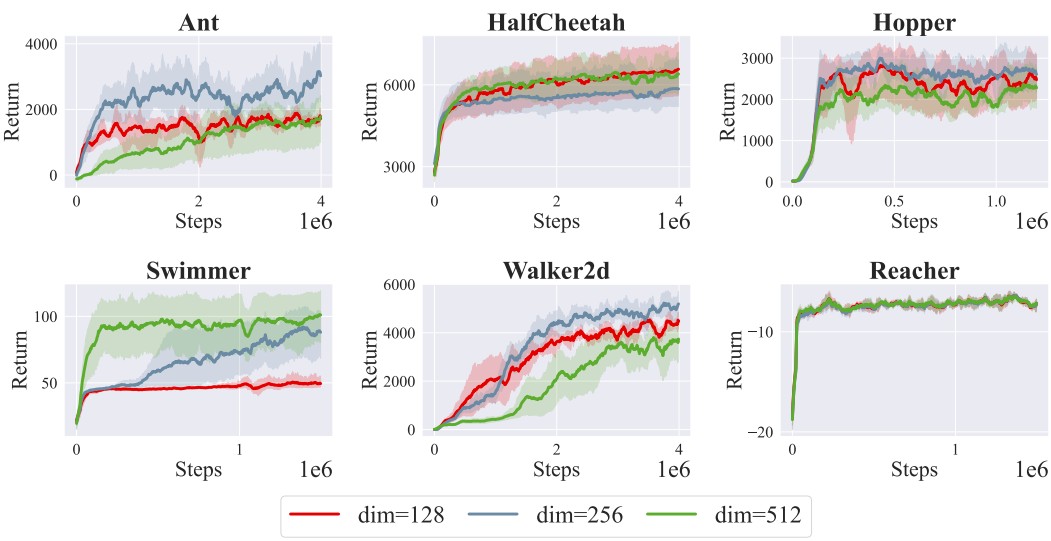

Figure 13: Comparison of DEER's performance with various dimensions and dropping probability 0.2.

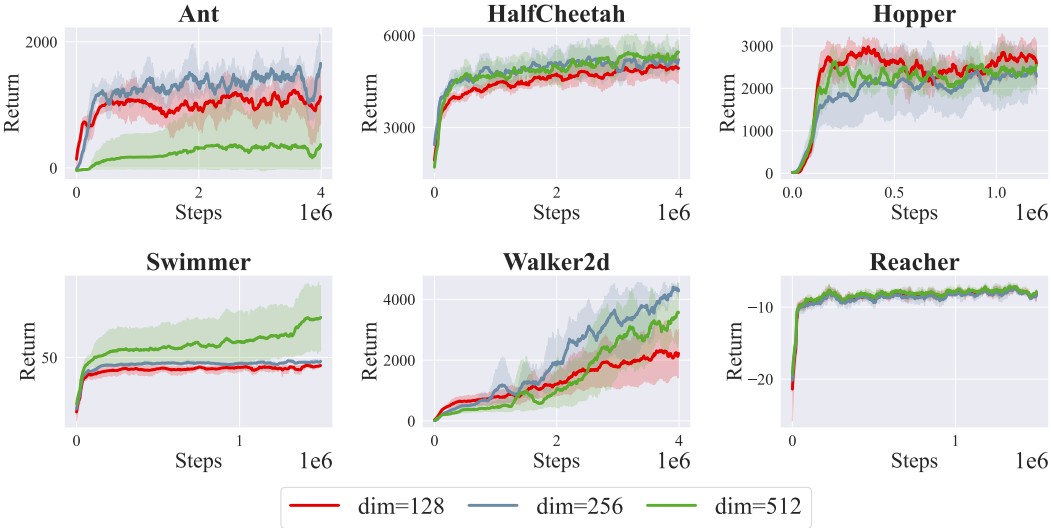

Figure 14: Comparison of DEER's performance with various dimensions and dropping probability 0.4.

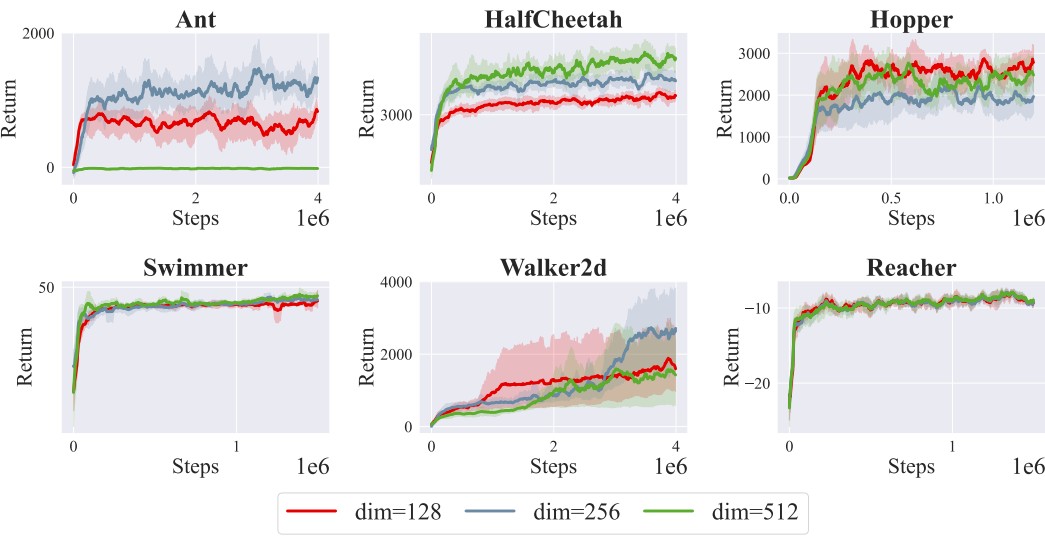

Figure 15: Comparison of DEER's performance with various dimensions and dropping probability 0.6.

## C.3 ABLATION STUDY

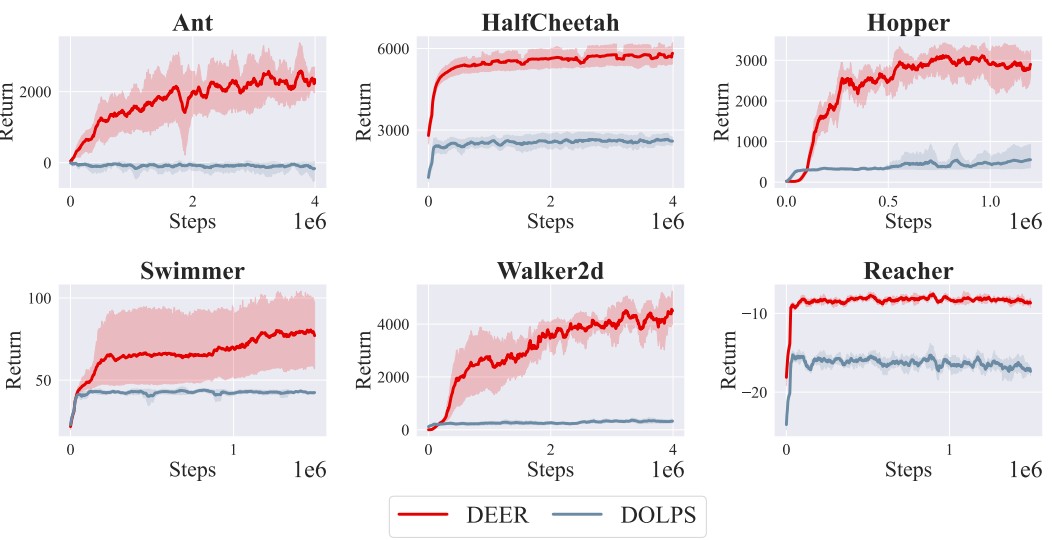

Figure 16: Comparison between DEER and DOLPS with a delay value of 4.

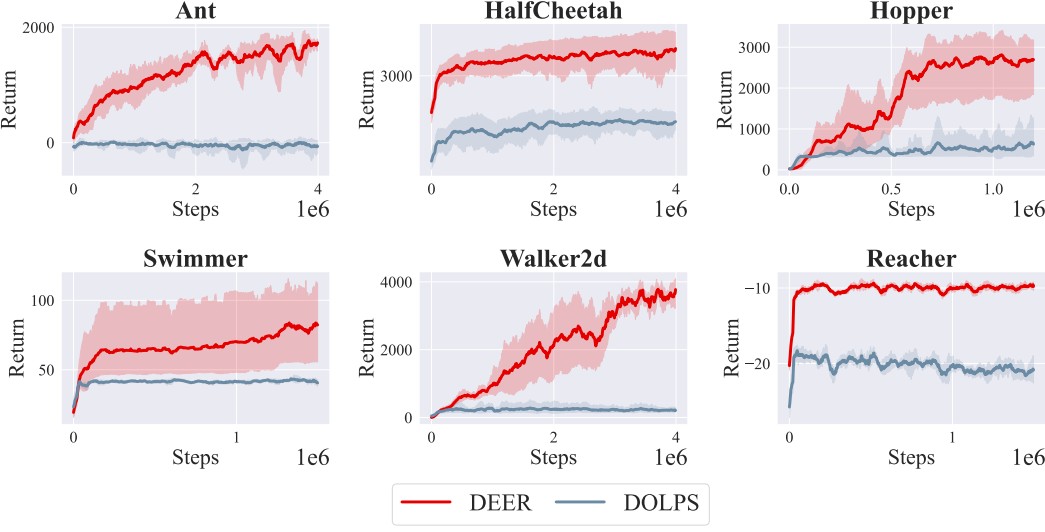

Figure 17: Comparison between DEER and DOLPS with a delay value of 6.

## C.4 MORE ANALYSIS ON DEER

**Content 1: time performance contrast between DEER and other algorithms.**

we conducted a comparative analysis of the time and performance of four algorithmsDEER, RLRD, DATS, and SACAS. Specifically, within the Hopper task, a fixed delay of 4 was established, with each algorithm undergoing three runs of 1 million environmental steps using different seeds. All algorithms were executed on the same NVIDIA GeForce RTX 4090 graphics card, and the time and performance comparison are illustrated in the Table 7.

In comparison to DEER and SACAS, RLRD and DATS incur excessive time consumption (despite RLRD displaying decent final performance). DEER requires more time compared to SACAS, attributed to both the pretraining duration and the encoding of information states into context representation. While DEER performs slightly lower than SACAS in the current settings, Figure 3 demonstrates that DEER surpasses SACAS, especially in the Swimmer task.

Table 7: Comparison of Time and Performance.

| Algorithm | Time(hour) | Scaled Return |
|---|---|---|
| DEER | 5.771±0.012 | 0.8±0.11 |
| RLRD | 70.67±0.4 | 0.85±0.05 |
| DATS | 81.2±0.087 | 0.61±0.13 |
| SACAS | 5.01±0.078 | 0.88±0.03 |

**Content 2: comparison between online and offline DEER.**

Our comparative experiments were exclusively performed on the Hopper and Walker2d tasks, each configured with a delay set at 4. The online version of DEER encompasses two primary modules: an encoder-decoder and a decision-making component. Initially, the encoder-decoder network is randomly initialized, and training data is gathered during real-time interactions within the environment. The encoder-decoder is updated every 300,000 steps. Regarding the decision-making aspect, the SAC framework is employed, wherein the replay buffer stores context representations derived from the encoder's information states. The entire training duration extends to 1 million steps. The comparative results are shown in the following Table 8. Comparing the results in the table, it's notably clear that the efficacy of offline DEER surpasses that of online DEER. This underscores the pivotal role played by a stable and well-pretrained encoder in determining the ultimate performance of the agent.

Table 8: Comparison of Online DEER and Offline DEER.

| Version of Deer | Online | Offline |
|---|---|---|
| Hopper | 751.75±623.66 | 2935.62±312 |
| Walker2d | 141±159.8 | 2623.14±542.8 |

**Content 3: comparison with state-of-art in Offline-to-Online RL.**

We undertook a comparison between the state-of-the-art algorithms in Offline-to-Online RL, namely, PEX(Zhang et al., 2023) and DEER. PEX's core approach involves initially learning a policy from an offline dataset, utilizing this learned policy as a candidate. Subsequently, another policy takes charge of further learning. Both policies interact with the environment in an adaptable manner. While training the offline policy, PEX requires reward information to guide its learning, in addition to states and actions. In contrast, DEER only relies on states and actions.

Within the MuJoCo environment, we conducted policy training using PEX across three continuous tasks (Ant, Hopper, and Walker2d), incorporating delays of 1, 2, 4, 6, and 8. Each scenario underwent experimentation using three different seeds. The offline policy training dataset for PEX aligns with that of DEER. The comparative experiment results are displayed in the Table 9. Analysis of the data in the table indicates that the final training outcome of PEX closely resembles that of an agent employing a random strategy. The subpar performance is likely associated with the utilized offline dataset, primarily constituted by a significant majority of random trajectories alongside a limited

number of expert trajectories. This composition within the offline data has resulted in PEX learning a strategy leaning towards randomness.

Table 9: Comparison of PEX and DEER

| Delay | 1 | | 2 | | 4 | | 6 | | 8 | |
|---|---|---|---|---|---|---|---|---|---|---|
| Algorithm | DEER | PEX | DEER | PEX | DEER | PEX | DEER | PEX | DEER | PEX |
| Ant | 4636 | 165 | 2903 | 111 | 2574 | 179.9 | 1653 | 314 | 1072 | 182 |
| Hopper | 2484 | 8.6 | 3096 | 6.9 | 2918 | 7.7 | 2565 | 6.5 | 2462 | 7.1 |
| Walker2d | 5411 | -5.1 | 4783 | -4.8 | 4119 | 1.6 | 3546 | -4.8 | 3074 | -4.5 |

**content 4: discussion on the impact of different context representation dimensions on agent performance.**

From our experimental results, mapping the information state to a higher-dimensional context representation resulted in better performance of the trained policy in delayed environments. Moreover, the improvement in performance does not have a direct proportional relationship with the increase in dimensions. In other words, higher dimensions do not necessarily yield better results. To clarify this conclusion further, we extended our experiments on the Hopper and Walker2d tasks by encoding the information state into dimensions similar to the original information state (32 and 64 dimensions) under a delay of 4. Subsequently, we employed the SAC framework and the same set of parameters for policy training. The final outcomes of the encoders with different dimensions on the Hopper and Walker2d tasks are presented in the table below.

Table 10: Comparison of different dimensions.

| Dimension | 32 | 64 | 128 | 256 | 512 |
|---|---|---|---|---|---|
| Hopper | $364\pm19$ | $2497\pm1012$ | $2994\pm325$ | $2988\pm432$ | $2414\pm562$ |
| Walker2d | $608\pm392$ | $728\pm123$ | $1646\pm286$ | $2290\pm969$ | $624\pm209$ |

Based on the data in the table above, it's evident that when the dimension of DEER's pre-trained encoder is too small, such as setting it to dim=32, the encoder's capability is too weak to effectively represent delay information within the information state, resulting in decreased performance. Conversely, when the dimension of DEER's pre-trained encoder is excessively large, for instance, with dim=512, although the encoder has numerous parameters and enhanced capabilities, there is a risk of potential overfitting to the current dataset, leading to decreased performance. Moreover, as the dimension shifts from 64 to 256, the performance improves for both tasks, indicating that the representation capability of the context representation indeed influences the final outcome.

**Content 5: comparison of different offline datasets.**

The reason behind structuring the offline dataset for pre-training the encoder to include a substantial amount of random trajectories and a few expert trajectories in the delay-free environment is to reflect a more realistic scenario, where random trajectories are predominant while expert trajectories are comparatively scarce. To analyze the impact of the offline dataset on the model, we introduced two additional types of offline datasets in the Hopper and Walker2d tasks, both with a delay of 4: one composed entirely of random trajectories and the other exclusively of expert trajectories. The subsequent policy training procedures align with DEER's methodology. The comparative outcomes from training with these three distinct offline datasets are presented in the table below.

Table 11: Comparison of different offline datasets

| Offline datasets | | Random | Large random and few expert | Expert |
|---|---|---|---|---|
| Tasks | Hopper | $2943.56\pm891.35$ | $2988.32\pm432.15$ | $3068.09\pm617.05$ |
| | Walker2d | $405.98\pm110.6$ | $2289.55\pm968.70$ | $510.67\pm95.25$ |

The experimental results from the Walker2d task in the Table 11 reveal an intriguing aspect: the quantity of expert trajectories doesn't directly correlate with superior final results. Essentially, the efficacy of DEER's pre-trained encoder seems to hinge closely on the distribution of state transitions within the provided dataset.

**Content 6: a fresh comparison in an alternative scenario of random delays.**

The random delay value $z_t$ is defined in the following manner:

$$z_t = \begin{cases} d_I, & \text{with probability} 1 - \mu, \\ z_{t-1} + 1, & \text{with probability } \mu \text{ and } z_{t-1} < d_I + d_M, \\ z_{t-1}, & \text{others.} \end{cases}$$

Based on this random delay setting, we compared four algorithms (DEER, RLRD, DATS, SACAS) in the Hopper and Walker2d tasks, setting $d_I = 2$, $d_M = 4$, and $\mu = 0.2, 0.4, 0.6$. Each algorithm was experimented with three different random seeds for 1 million environment steps in each environment configuration. The comparative results are shown in Table12 - Table14.

Table 12: Comparison of algorithms in a drop rate of 0.2.

| Tasks($\mu = 0.2$) | DEER | RLRD | DATS | SACAS |
|---|---|---|---|---|
| Hopper | 2583.90±915.03 | 2391.16±473.86 | -1064.50±426.87 | 820.94±15.58 |
| Walker2d | 3940.05±1539.67 | 3013.65±571.69 | -1196.97±418.48 | 670.84±74.90 |

Table 13: Comparison of algorithms in a drop rate of 0.4.

| Tasks($\mu = 0.4$) | DEER | RLRD | DATS | SACAS |
|---|---|---|---|---|
| Hopper | 2542.54±960.73 | 1609.96±403.12 | -754.47±225.30 | 609.33±15.80 |
| Walker2d | 4247.60±1441.24 | 2062.80±198.73 | -1237.48±350.60 | 518.13±47.12 |

Table 14: Comparison of algorithms in a drop rate of 0.6.

| Tasks($\mu = 0.6$) | DEER | RLRD | DATS | SACAS |
|---|---|---|---|---|
| Hopper | 2129.59±995.17 | 638.93±230.15 | -735.94±197.38 | 398.93±48.83 |
| Walker2d | 2578.52±1614.18 | 908.23±222.47 | -672.75±238.66 | 360.65±28.46 |

From the table above, it's evident that even under the new random delay settings, the performance of DEER significantly surpasses other algorithms (although there might be instances where it closely aligns with RLRD scores). This underscores the significance of state representation in decision-making, further emphasizing that the pre-trained encoder in the DEER algorithm indeed extracts delay-related information from the information state.

C.5 LIMITATION

The encoder in DEER plays a critical role in extracting informative features from delayed state and action data, enabling the agent to effectively tackle both constant and random delays issues. Consequently, the quality of the encoder has a substantial impact on the agents overall performance.

The difference in the number and type of trajectories used to pretrain the encoder, as shown in Table 5, can be explained as follows: Firstly, the variation in the number of random trajectories across different tasks is aimed at ensuring that the training set for each task's encoder contains a comparable number of transitions to what is required for training the agent to achieve expert-level performance in an undelayed environment. Moreover, trajectories generated by random policies primarily demonstrate the initial state distribution of the environment and show partial state transition functions, while expert trajectories exhibit the opposite characteristics. DEER achieves satisfactory performance with just 10 expert trajectories in all tasks apart from the Walker, as shown in

Table 5. Figure 18 illustrates the performance comparison of DEER in Walker2d with 10 and 60 expert trajectories. It is evident that a greater number of expert trajectories leads to improved final performance by the agent.

In summary, when using DEER to address a new task with delays, it is advisable to provide the encoder with a dataset of transitions that is comparable to the training data used to achieve expert-level performance in a delay-free environment, in addition to incorporating as many expert trajectories as possible.

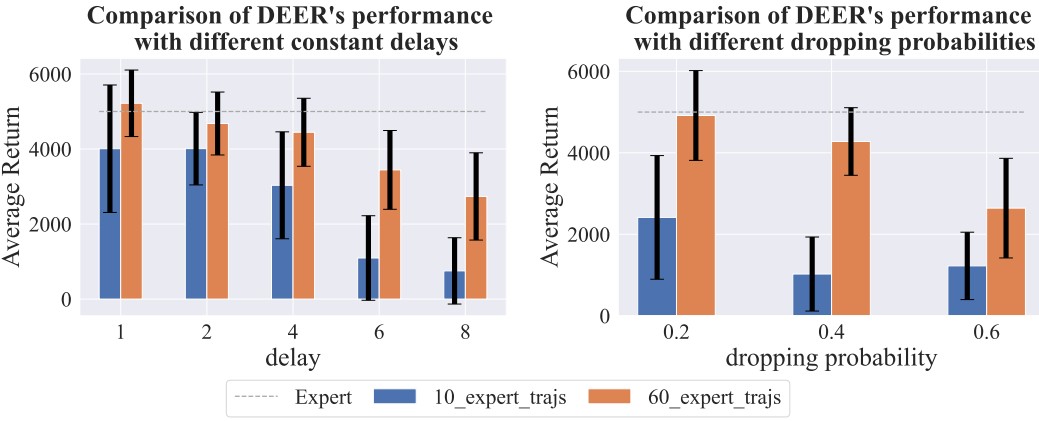

Figure 18: Comparison of DEER's performance with different number of expert trajectories in Walker2d.

