# OpenReview forum: "DEER: A Delay-Resilient Framework for Reinforcement Learning with Variable Delays"
_ICLR.cc/2024/Conference — Submitted to ICLR 2024_

### Official Review · Reviewer_yS4B · 2023-10-31

**Soundness:** 3 good
**Presentation:** 2 fair
**Contribution:** 2 fair
**Rating:** 6
**Confidence:** 3

**Summary:**

This paper focuses on reinforcement learning in a Markov decision processes in which the agent observes its current state only after a delay of random length. The proposed algorithm DEER uses offline data to learn an embedding for the delayed state and its corresponding action sequence (s_{t-d}, a_{t-d}, …, a_t). This embedding enables the learning agent to generalize to environments with different or randomized delays. Empirically, DEER outperforms baselines in MuJoCo benchmark tasks with constant and randomized delays.

**Strengths:**

1. The problem setting (MDP with delayed state feedback) is especially relevant to real-world RL tasks.
1. DEER is quite versatile and can be used with any RL algorithm.

**Weaknesses:**

1. It's unclear why breaking the end-to-end decision process into an encoding phase and a decision-making phase increases interpretability. I don't believe this claim is supported by the current experiments. The paper should clarify this contribution.
1. Since DEER essentially learns a dynamics model that predicts a sequence of states given an initial state and sequence of actions, my understanding is that DEER still suffers the same drawbacks as model-based approaches.

1. DEER seems to have an unfair advantage over the baselines in Figures 4 and 5. In particular, DEER assumes access to an offline dataset, but the baselines do not. I believe DEER could in principle train its encoder in an online fashion -- if so, then an online variant of DEER would provide a fairer comparison across algorithms.

**Other Comments:

1. Figure 1 and 3 are essentially the same. I suggest replacing Figure 1 with Figure 3, and then include the “undelayed trajectory buffer” icon from figure 1.

2. Advantage (1) seems to subsume advantage (2). I think it's sufficient to say DEER enables agents to generalize to diverse delay environments.

3. In the abstract and conclusion, it is unclear what “tasks involving delays” and “delay issues” mean. These phrases are immediately clear from the introduction, but some researchers (such as myself) read the abstract and conclusion before reading the full paper. I suggest clarifying that you’re referring to when the learning agent receives its next observation after taking an action.

4. The paper should explicitly define "augmented states. I believe "augmented state" refers the concatenation of state and action information, but this term makes me think of a synthetic data generated by a data augmentation function (e.g. a random cropping and recoloration of an image state).

5. GRU not defined at its first occurrence.

6. The “Offline assisted Online RL” section would be more readable if you used separate paragraphs for (1) and (2).

7. It would help readers if you briefly stated how Seq2Seq effectively handles delays.

10. Last paragraph at section 4 is an experimental detail and would fit better at beginning of paragraph 2 in section.

**Questions:**

1. First line of page 2: What does “dynamic environments” mean?

1. What the difference between d_M and D? The seem to represent the same quantity.

1. The paper mentions the computation drawback of existing approaches. How does DEER compare computationally to the baselines considered?

1. How does breaking the end-to-end decision-making process into two stages improve interpretability? How is DEER more interpretable than say RLRD and DATS?

1. Does the encoder increase or decrease the dimensionality of the augmented state (s_{t-d}, a_{t-d}, …, a_t)? Does DEER intend to *compress* the MDP?

---

> ### Author Response · Authors · 2023-11-21
> **Response to Reviewer yS4B(Part 1/3)**
>
> We are glad that the reviewer found our work is especially relevant to real-world RL tasks and can be used with any RL algorithm. We hope to provide clarification to address the concerned issues. For the convenience of the reviewer, we reiterate the comments below.
>
> >**[Q1]**  It's unclear why breaking the end-to-end decision process into an encoding phase and a decision-making phase increases interpretability. I don't believe this claim is supported by the current experiments. The paper should clarify this contribution.
>
> **[A1]**  Information states consist of the most recently observed states and their corresponding action sequences. The SACAS method utilizes information states as decision states to design value and policy functions. When there is significant delay, the dimensionality of the action part within the information states can be much larger than that of the states themselves. The SACAS method operates as a black-box process, disregarding the mentioned issue and completing training end-to-end, with the quality depending on the neural network's capacity and parameter settings. However, in the initial phase of our experiments, the delayed state information corresponding to delayed actions is actually known. Using this information, an encoder-decoder can map the information states into a latent space. This process, unlike SACAS's end-to-end random exploration of network parameters, is more directive, compelling the model to start from a good state representation. The improved interpretability is reflected here.
>
>
> >**[Q2]**  Since DEER essentially learns a dynamics model that predicts a sequence of states given an initial state and sequence of actions, my understanding is that DEER still suffers the same drawbacks as model-based approaches.
>
> **[A2]** Model-based algorithms typically involve constructing a dynamics model through real-time interactions with the environment, followed by planning using this model. Consequently, these algorithms are often vulnerable to model and iteration errors. DEER, however, leverages existing data from delay-free environments to learn an encoder-decoder for multi-step prediction. During the decision-making phase, DEER employs the encoder's representation of input states and action sequences, termed as context representation, rather than the final state from multi-step predictions. This representation extracts decision-relevant information from the most recent observed states and action sequences, thereby mitigating sensitivity to deviations from the true state caused by model and iteration errors.
>
> >**[Q3]**  DEER seems to have an unfair advantage over the baselines in Figures 4 and 5. In particular, DEER assumes access to an offline dataset, but the baselines do not. I believe DEER could in principle train its encoder in an online fashion -- if so, then an online variant of DEER would provide a fairer comparison across algorithms.
>
> **[A3]** Due to constraints in computational resources and time, our comparative experiments were exclusively performed on the Hopper and Walker2d tasks, each configured with a delay set at 4. The online version of DEER encompasses two primary modules: an encoder-decoder and a decision-making component. Initially, the encoder-decoder network is randomly initialized, and training data is gathered during real-time interactions within the environment. The encoder-decoder is updated every 300,000 steps. Regarding the decision-making aspect, the SAC framework is employed, wherein the replay buffer stores context representations derived from the encoder's information states. The entire training duration extends to 1 million steps. The comparative results are shown in the following table:
>
> | Version of Deer | Online        | Offline       |
> |-----------------|---------------|---------------|
> | Hopper          | 751.75±623.66 | 2935.62±312   |
> | Walker2d        | 141±159.8     | 2623.14±542.8 |
>
>
> >**[Q4]** The paper should explicitly define "augmented states. I believe "augmented state" refers the concatenation of state and action information, but this term makes me think of a synthetic data generated by a data augmentation function (e.g. a random cropping and recoloration of an image state).
>
> **[A4]**  Thanks for the suggestion. We replaced all "augmented states" with "information states".

---

> ### Author Response · Authors · 2023-11-21
> **Response to Reviewer yS4B(Part 2/3)**
>
> >**[Q5]** It would help readers if you briefly stated how Seq2Seq effectively handles delays.
>
> **[A5]**  For addressing the delay issue, the Seq2Seq model in the paper serves two purposes: encoder pretraining and policy training. In the encoder pretraining phase, the dataset from the delay-free environment is initially randomly reconstructed to create information state sets with various delay sizes and their corresponding delayed state sequence sets. Specifically, assuming time $t$ and delay $d$ where $d \in [d_I, d_I+d_M]$, with $D=d_I+d_M$, the information state is denoted as $I_t = (s_{t-d}, a_{t-d},a_{t-d+1}, \cdots, a_{t-1})$，and the delayed state sequence is $g_t=(s_{t-d+1}, \cdots, s_{t-1})$.  For information states with a delay smaller than $D$, to facilitate batch training, these information states are zero-padded until their length reaches $D$. Next, a Seq2Seq encoder-decoder model is employed to predict the state based on the input information state and output the delayed state sequence. Specifically, a multi-layer perceptron encodes each element of the information state to generate corresponding embeddings. These embeddings are then fed into a GRU module to generate a hidden feature vector of a dimension determined by hyper-parameters. Finally, the Seq2Seq model is optimized using MSE loss to improve the accuracy of state sequence prediction, thereby refining the context representation of the information states. These hidden states are the ones utilized for policy learning.
> During the policy training phase, the agent interacts with the environment in real-time, causing the length of the information state $I_t$ to dynamically change due to delays. We employ the pretrained Seq2Seq encoder, where information states of varying lengths $I_t$ are fed into the encoder. After several iterations, the encoder extracts delayed state information and maps it to fixed-dimensional context representations. Subsequent policy training relies on these context representations rather than the original state space. Moreover, there is no need for additional adjustments to the chosen policy algorithm due to changes in delays.
>
> >**[Q6]** First line of page 2: What does “dynamic environments” mean?
>
> **[A6]** “dynamic environments” means non-deterministic environments. The corresponding sentence is revised as: While effective in deterministic environments, their adaptability in nondeterministic environments needs further improvement.
>
> >**[Q7]** What the difference between d_M and D? The seem to represent the same quantity.
>
> **[A7]** $d_I \in Z^{+}$ denotes the intrinsic delay value resulting from long-distance transmission or heavy data transfers. Meanwhile, $d_M \in Z^{+}$ represents the maximum number of additional dropping steps, ensuring that $d_I + d_M$ remains within the agent’s capacity. The sum of these values is defined as $D = d_I + d_M$.
>
> >**[Q8]** The paper mentions the computation drawback of existing approaches. How does DEER compare computationally to the baselines considered?
>
> **[A8]** To address this inquiry, we conducted a comparative analysis of the time and performance of four algorithms—DEER, RLRD, DATS, and SACAS. Specifically, within the Hopper task, a fixed delay of 4 was established, with each algorithm undergoing three runs of 1 million environmental steps using different seeds. All algorithms were executed on the same NVIDIA GeForce RTX 4090 graphics card, and the time and performance comparison are illustrated in the Table.
> | Algorithm | Time(hour)  | Scaled Return |
> |-----------|-------------|---------------|
> | DEER      | 5.771±0.012 | 0.8±0.11      |
> | RLRD      | 70.67±0.4   | 0.85±0.05     |
> | DATS      | 81.2±0.087  | 0.61±0.13     |
> | SACAS     | 5.01±0.078  | 0.88±0.03     |
>
> >**[Q9]** Modification in paper: figure 1 and 3 are essentially the same, advantage (1) seems to subsume advantage (2), the meaning of tasks involving delays” and “delay issues” in the Introduction,  making “ Offline assisted Online RL” section more readable, GRU not defined at its first occurrence and last paragraph at section 4 would fit better at beginning of paragraph 2 in section.
>
> **[A9]** The aforementioned content will be updated in my newly uploaded PDF.

---

> > ### Author Response · Authors · 2023-11-21
> > **Response to Reviewer yS4B(Part 3/3)**
> >
> > >**[Q10]** Does the encoder increase or decrease the dimensionality of the augmented state $(s_{t-d}, a_{t-d}, …, a_t)$? Does DEER intend to compress the MDP?
> >
> > **[A10]** From our experimental results, mapping the information state to a higher-dimensional context representation resulted in better performance of the trained policy in delayed environments. Moreover, the improvement in performance does not have a direct proportional relationship with the increase in dimensions. In other words, higher dimensions do not necessarily yield better results. To clarify this conclusion further, we extended our experiments on the Hopper and Walker2d tasks by encoding the information state into dimensions similar to the original information state (32 and 64 dimensions) under a delay of 4. Subsequently, we employed the SAC framework and the same set of parameters for policy training. The final outcomes of the encoders with different dimensions on the Hopper and Walker2d tasks are presented in the table below.
> >
> > | Dimension | 32            | 64              | 128            | 256            | 512            |
> > |-----------|---------------|-----------------|----------------|----------------|----------------|
> > | Hopper    | 363.99±19.15  | 2496.68±1011.77 | 2993.60±325.07 | 2988.32±432.15 | 2413.60±562.31 |
> > | Walker2d  | 607.68±392.12 | 728.15±122.54   | 1645.92±286.45 | 2289.55±968.70 | 623.75±209.45  |
> >
> > Based on the data in the table above, it's evident that when the dimension of DEER's pre-trained encoder is too small, such as setting it to dim=32, the encoder's capability is too weak to effectively represent delay information within the information state, resulting in decreased performance. Conversely, when the dimension of DEER's pre-trained encoder is excessively large, for instance, with dim=512, although the encoder has numerous parameters and enhanced capabilities, there is a risk of potential overfitting to the current dataset, leading to decreased performance. Moreover, as the dimension shifts from 64 to 256, the performance improves for both tasks, indicating that the representation capability of the context representation indeed influences the final outcome.

---

> ### Author Response · Authors · 2023-11-22
>
> Has our response sufficiently addressed your inquiries? Are there any additional problems requiring discussion? We eagerly anticipate further correspondence with you. Please let us know if there are any additional comments, questions, or concerns. Many thanks again for the invaluable hard work of all reviewers to improve our work.

---

> > ### Comment · Reviewer_yS4B · 2023-11-22
> >
> > The you for the clarifications; my and comments questions have been addressed. After reading through reviewers comments and the authors' response, I've decided to raise my score.

---

### Official Review · Reviewer_pa1n · 2023-11-01

**Soundness:** 2 fair
**Presentation:** 3 good
**Contribution:** 2 fair
**Rating:** 5
**Confidence:** 3

**Summary:**

This paper addresses the challenge of delays in reinforcement learning (RL) by proposing a framework, DEER (Delay-resilient Encoder-Enhanced RL), aiming to enhance interpretability and overcome random delay issues. It employs a pretrained encoder to convert delayed states and variable-length action sequences into hidden states, providing delay-free states for policy training. Empirical experiments are performed in various delayed scenarios on Gym and Mujoco to demonstrate the performance of DEER against existing RL algorithms in both constant and random delays.

**Strengths:**

Unlike the existing methods that address delays in RL by performing state augmentation, this paper introduces a novel framework, namely, deer, to employ a pretrained encoder to encode delayed states along with their corresponding action sequences for further policy training. Delay is an important challenge facing RL in practice, the setting that this paper studies is of practical concern. To address this issue, the authors introduce Delay-resilient Encoder-Enhanced RL (DEER) which leverages an encoder pretrained on undelayed offline datasets to enhance online learning in delayed environments. This approach offers advantages such as generalizability to diverse delay environments, versatility in addressing constant and random delays without modifying the agent's structure, and improved interpretability by breaking the decision process into distinct stages. In particular,
1. The proposed framework is able to integrate seamlessly with standard RL algorithms without additional modifications.
2. It is effective in both constant and random delay environments, eliminating the necessity for prior knowledge of delay characteristics.
3. Empirical studies show that equipped with SAC, it enjoys competitive efficiency and performance compared to existing state-of-the-art methods on Gym and Mujoco tasks.

**Weaknesses:**

The proposed method relies on encoding the augmented state with embeddings. When the state space is large and high-dimensional, further augmentation can be computationally inefficient. Authors should clarify whether higher-dimension embeddings are necessary as a context representation to achieve the desired performance. It is also desirable to have experiments that assess how the performance (w.r.t. return) is affected by the size of embeddings, and whether higher-dimension yields significantly better performance. When a compact / low-dimensional embedding is used, can it still generalize well to different delay scenarios.

The current method is a direct application of the encoder-decoder framework based on Seq2Seq scheme. Algorithmically, the novelty is limited. The authors are suggested to highlight what are the technical novelties in their algorithmic framework.

**Questions:**

The authors utilize an undelayed offline dataset to pretrain the encoder. In this case, how can it capture the delays in the embeddings of the states when deployed to the delayed environments?

---

> ### Author Response · Authors · 2023-11-20
> **Response to Reviewer pa1n (Part 1/2)**
>
> We thank the reviewer for the useful and detailed comments. In the following, we aim to address your questions and concerns. For the convenience of the reviewer, we reiterate the comments below.
>
> >**[Q1]**  The proposed method relies on encoding the augmented state with embeddings. When the state space is large and high-dimensional, further augmentation can be computationally inefficient. Authors should clarify whether higher-dimension embeddings are necessary as a context representation to achieve the desired performance. It is also desirable to have experiments that assess how the performance (w.r.t. return) is affected by the size of embeddings, and whether higher-dimension yields significantly better performance. When a compact / low-dimensional embedding is used, can it still generalize well to different delay scenarios.
>
> **[A1]** From our experimental results, mapping the information state to a higher-dimensional context representation resulted in better performance of the trained policy in delayed environments. Moreover, the improvement in performance does not have a direct proportional relationship with the increase in dimensions. In other words, higher dimensions do not necessarily yield better results. To clarify this conclusion further, we extended our experiments on the Hopper and Walker2d tasks by encoding the information state into dimensions similar to the original information state (32 and 64 dimensions) under a delay of 4. Subsequently, we employed the SAC framework and the same set of parameters for policy training. The final outcomes of the encoders with different dimensions on the Hopper and Walker2d tasks are presented in the table below.
>
> | Dimension | 32            | 64              | 128            | 256            | 512            |
> |-----------|---------------|-----------------|----------------|----------------|----------------|
> | Hopper    | 363.99±19.15  | 2496.68±1011.77 | 2993.60±325.07 | 2988.32±432.15 | 2413.60±562.31 |
> | Walker2d  | 607.68±392.12 | 728.15±122.54   | 1645.92±286.45 | 2289.55±968.70 | 623.75±209.45  |
>
> Based on the data in the table above, it's evident that when the dimension of DEER's pre-trained encoder is too small, such as setting it to dim=32, the encoder's capability is too weak to effectively represent delay information within the information state, resulting in decreased performance. Conversely, when the dimension of DEER's pre-trained encoder is excessively large, for instance, with dim=512, although the encoder has numerous parameters and enhanced capabilities, there is a risk of potential overfitting to the current dataset, leading to decreased performance. Moreover, as the dimension shifts from 64 to 256, the performance improves for both tasks, indicating that the representation capability of the context representation indeed influences the final outcome.
>
> >**[Q2]** The current method is a direct application of the encoder-decoder framework based on Seq2Seq scheme. Algorithmically, the novelty is limited. The authors are suggested to highlight what are the technical novelties in their algorithmic framework.
>
> **[A2]** The technical novelties in DEER are as follows:
> Firstly, DEER stands as the pioneering approach that bridges data from delay-free tasks with applications in delayed tasks. Secondly, DEER doesn't necessitate modifications to subsequent policy algorithms; it seamlessly applies existing algorithms. Lastly, although DEER utilizes a simple Seq2Seq model, context representation derived from its encoder effectively extracts delay-related information, significantly bolstering decision-making capabilities.

---

> ### Author Response · Authors · 2023-11-20
> **Response to Reviewer pa1n (Part 2/2)**
>
> >**[Q3]** The authors utilize an undelayed offline dataset to pretrain the encoder. In this case, how can it capture the delays in the embeddings of the states when deployed to the delayed environments?
>
> **[A3]** The paper only discusses communication delay within the tasks, where the misalignment between the observed states by the agent and the executed actions leads to the current delay environment. We transform delay-free offline datasets to match different delay sizes within the delayed environment. Subsequently, we employ a Seq2Seq encoder-decoder model to predict states in an autoregressive manner. The encoded information state, referred to as the context representation, is then used as the decision-making state. In the delayed environment, at time $t$, the information state is defined as $ I_t = (s_{t-d}, a_{t-d},a_{t-d+1}, \cdots, a_{t-1}) $ with the corresponding representation as $c_t$, and the decision action as $a_t$. If the subsequent time $t+1$ observes the state $s_{t-d+1}$, the information state updates to $I_{t+1} = (s_{t-d+1}, a_{t-d+1},a_{t-d+2}, \cdots, a_{t})$ with the corresponding representation as $c_{t+1}$. However, if the state s_{t-d+1} is not observed, the information state updates to $ I_{t+1} = (s_{t-d}, a_{t-d},a_{t-d+1}, \cdots, a_{t}) $ with the corresponding representation as $c_{t+1}^{‘}$, and the decision action as $a_{t+1}$.

---

> ### Author Response · Authors · 2023-11-22
>
> Has our response sufficiently addressed your inquiries? Are there any additional problems requiring discussion? We eagerly anticipate further correspondence with you. Please let us know if there are any additional comments, questions, or concerns. Many thanks again for the invaluable hard work of all reviewers to improve our work.

---

> > ### Author Response · Authors · 2023-11-23
> >
> > In [A1], we conducted an analysis of how various dimensions affect the final performance of DEER. [A2] expounded on the technical innovations within the algorithm framework. [A3] clarified how DEER handles different delays within a delayed environment. We sincerely look forward to engaging in further discussions with you.

---

> ### Author Response · Authors · 2023-11-23
>
> We hope our response has satisfactorily addressed your queries and concerns. Please do not hesitate to reach out if you have any further questions or require additional clarification. We are truly grateful for the invaluable contributions made by the reviewer, which have significantly enhanced the quality of our work.

---

> > ### Comment · Reviewer_pa1n · 2023-12-02
> >
> > Dear authors,
> >
> > Thanks for your detailed replies and further information! After reading the replies, I am still hesitant regarding the algorithmic novelty. Considering other reviewers' opinions and further responses, I am inclined to maintain my original assessment. The manuscript can be further polished based on the comments.

---

### Official Review · Reviewer_HvM3 · 2023-11-01

**Soundness:** 3 good
**Presentation:** 3 good
**Contribution:** 3 good
**Rating:** 5
**Confidence:** 5

**Summary:**

This paper proposed Delay-resilient Encoder-Enhanced RL (DEER), a framework to handle delayed observations in deep RL. DEER leverages an encoder model to interpret delayed observations and actions into a latent representation space using delay-free offline experience, which aims to enhance the robustness of the encoding agains random, variant delay steps. The encoder is empowered with the Seq2Seq model by predicting delay-free observations, and the encoded latent features are used as the input for downstream model-free RL policy and critic network. The authors performed experiments on MuJoCo robotic movement tasks, where DEER outperforms previous methods. Ablation studies were also conducted. In sum, this work appears as a intereting attemptation to mitigated the difficulties in delayed RL task.


**[Post-rebuttal]**
I appreciate the author's repsonses. However, my concerns have not been fully addressed. E.g., there is no reply to my last comment in Weakness, and my inquiry to evaluate DEER's performance in delay=0 has not been addressed by the authors (for my comment "DEER should also be compared with vanilla SAC in delay=0 case"). Also, it is still not explained why DEER outperforms expert very much in the Swimmer task.

Overall, I decide to maintain my original rating and I believe a comprehensive revision of the paper will be helpful to improve this work.

**Strengths:**

- The delay problem being address in this work is important while under-explored. This work is a good attempt.
- DEER performs well and beats existing methods in delayed RL problems.
- The entire paper is generally well-written and easy to follow.
- The authors performed reasonable ablation studies to address the effect of key hyper-parameters design choice

**Weaknesses:**

- The novelty of DEER is moderate. From my understanding, the key novelty is a framework combining "offline-assisted online RL" and "Encoder in RL".

- The definition of RDDMP is quite complex and probably unrealistic. One concern is about the random delay value $z_t$ with dropping probablity $\mu$. $z_t = d_I$ if $z_{t-1} = d_I + d_M$, which means the $d_M$ steps of delay suddenly "disappear".  How can this happen in reality? My suggestion is to modify it as  (this ensures that there is no sudden change of delay steps)

   - $z_t = z_{t-1} - 1$ if $z_{t-1} > d_I$ and with probability $ \mu_{catch} $
   - $z_t = z_{t-1} + 1$ if $z_{t-1} < d_I + d_M$ and with probability $ \mu_{lag}$
   - $z_t = z_{t-1}$  otherwise

- More implementation details should be mentioned for the baseline models (RLRD, DATS, SACAS). See the questions below.

- DEER should also be compared with vanilla SAC in delay=0 case. This is important to understand the effect of pretrained encoder in deep RL and probably explain why DEER significantly outforms expert in Swimmer (Figure 4).

- While the authos commented three main contributions in the end of Introduction, they seems the same thing for me -- "We propose a framework which enhances agent performance in delayed RL tasks".

## Minor

- Fonts are too small in Figure 3. I cannot recognize the subscripts on the printed paper.
- Figure 6, 7 etc. could be more clear with different linestyles in addition to colors for color-blind-friendly visualization.
- A table summarizing the numerical reults (including the converged task return of each task and average performance of all tasks) could help the readers get easier access to the empirical results.
- A typo in the title "resillient"

**Questions:**

- How did the authos obtain the result of the baseline models? E.g., DATS used less training samples (environmental steps) than 1e6 in their paper. Also, How many steps of actions are augmented in the state in SACAS? What are the hyperparameters and how did the authors selected them?

- In the end of the first paragraph, it was mentioned minor delays markedly amplifies the risk in self-driving. Is there any reference? Following this question, are there more examples (better with data) of practical scenarios that suffers heavily from delays?

- Why the performance of DEER is sensitive to dimension of latent representation? In particular, why dim=512 results in significantly worse performance in Walker2D (Figure 6)? My expectation is large dimension should work better or similar. A potentially related work is (Ota et al. 2020)

Overall, the being touched problem, i.e., delay in deep RL is a quite important problem and the efforts toward solving this problem should be valuable contribution to the deep learning and robotics community. While there are some tacts in the current version of the manuscript. I am happy to discuss with the authors and will increase my score if my concerns are well addressed.


### REF
- Ota K, Oiki T, Jha D, et al. Can increasing input dimensionality improve deep reinforcement learning?[C]//International conference on machine learning. PMLR, 2020: 7424-7433.

---

> ### Author Response · Authors · 2023-11-20
> **Response to  Reviewer HvM3 (Part 1/3)**
>
> We are glad that the reviewer found our work important and thoght it was a good attempt. We hope to provide clarification to address the concerned issues. For the convenience of the reviewer, we reiterate the comments below.
>
> >**[Q1]**  The novelty of DEER is moderate. From my understanding, the key novelty is a framework combining "offline-assisted online RL" and "Encoder in RL".
>
> **[A1]** The reviewer has indeed grasped the core concept of the DEER method. However, summarizing DEER as a mere fusion of "offline-assisted online RL" and "Encoder in RL" is somewhat simplistic. Firstly, DEER stands as the pioneering approach that bridges data from delay-free tasks with applications in delayed tasks. Secondly, DEER doesn't necessitate modifications to subsequent policy algorithms; it seamlessly applies existing algorithms. Lastly, although DEER utilizes a simple Seq2Seq model, context representation derived from its encoder  effectively extracts delay-related information, significantly bolstering decision-making capabilities.
>
> >**[Q2]** The definition of RDDMP is quite complex and probably unrealistic. One concern is about the random delay value $z_t$  with dropping probablity $\mu$. $z_t=d_I$ if $z_{t-1}=d_I+d_M$. which means the $d_M$ steps of delay suddenly "disappear". How can this happen in reality? My suggestion is to modify it in another formation.
>
> **[A2]** The reviewer's statement is accurate. In our defined scenario, when the cumulative count of consecutive frame drops reaches $d_M$, it resets. There are two perspectives to explain this setup:
> 1.The probability of dropping frames, $\mu$, for each instance, compounded over $d_M$ consecutive drops, equates to $d_M^{\mu}$. This value tends to be relatively small, signifying that it is an infrequent occurrence.
> 2.In real-world scenarios, when an agent encounters $d_M$ consecutive instances of not receiving a state, it surpasses the agent's capacity to handle delay. Hence, it necessitates the remote end to ensure the appearance of the next state.
> The proposed scenario by the reviewer is indeed more reasonable but requires a slight modification: when the agent observes the returned state, the current delay reverts to $d_I$, representing the intrinsic delay value resulting from long-distance transmission or heavy data transfers. Hence, $z_t = d_I$ if $z_{t-1} > d_I$ with a probability of $\mu_{catch}$. Based on this stochastic delay setting, we compared four algorithms (DEER, RLRD, DATS, SACAS) in the Hopper and Walker2d tasks, setting $d_I=2$, $d_M=4$, and $\mu=0.2, 0.4, 0.6$. Each algorithm was experimented with three different random seeds for 1 million environment steps in each environment configuration. The comparative results are shown below.
>
> | Tasks($\mu=0.2$) | DEER            | RLRD           | DATS            | SACAS        |
> |----------------|-----------------|----------------|-----------------|--------------|
> | Hopper         | 2583.90±915.03  | 2391.16±473.86 | -1064.50±426.87 | 820.94±15.58 |
> | Walker2d       | 3940.05±1539.67 | 3013.65±571.69 | -1196.97±418.48 | 670.84±74.90 |
> |                |                 |                |                 |              |
>
> | Tasks($\mu=0.4$) | DEER            | RLRD           | DATS            | SACAS        |
> |----------------|-----------------|----------------|-----------------|--------------|
> | Hopper         | 2542.54±960.73  | 1609.96±403.12 | -754.47±225.30  | 609.33±15.80 |
> | Walker2d       | 4247.60±1441.24 | 2062.80±198.73 | -1237.48±350.60 | 518.13±47.12 |
> |                |                 |                |                 |              |
>
> | Tasks($\mu=0.6$) | DEER            | RLRD          | DATS           | SACAS        |
> |----------------|-----------------|---------------|----------------|--------------|
> | Hopper         | 2129.59±995.17  | 638.93±230.15 | -735.94±197.38 | 398.93±48.83 |
> | Walker2d       | 2578.52±1614.18 | 908.23±222.47 | -672.75±238.66 | 360.65±28.46 |
> |                |                 |               |                |              |
>
> From the table above, it's evident that even under the requested random delay settings by the reviewer, the performance of DEER significantly surpasses other algorithms (although there might be instances where it closely aligns with RLRD scores). This underscores the significance of state representation in decision-making, further emphasizing that the pre-trained encoder in the DEER algorithm indeed extracts delay-related information from the information state.
>
>
> >**[Q3]** DEER should also be compared with vanilla SAC in delay=0 case. This is important to understand the effect of pretrained encoder in deep RL and probably explain why DEER significantly outforms expert in Swimmer.
>
> **[A3]** Whether in fixed or random delay settings, the legend labeled "Expert" in the graph corresponds to the performance of vanilla SAC trained with delay=0, as depicted in the Figure 4 and Figure 5 .

---

> ### Author Response · Authors · 2023-11-20
> **Response to Reviewer HvM3 (Part 2/3)**
>
> >**[Q4]**  How did the authos obtain the result of the baseline models? E.g., DATS used less training samples (environmental steps) than 1e6 in their paper. Also, How many steps of actions are augmented in the state in SACAS? What are the hyperparameters and how did the authors selected them?
>
> **[A4]** DATS uses fewer training samples, but during training, the probabilistic ensemble network in the dynamics part and the sampling method in the planning part can lead to an increase in the wall-clock time. Moreover, from the results of experiments with random delays, there is a moderate improvement in the final performance with increased steps in the experiments. The code repository for DATS can be found at [1], using default parameters. However, to reach 1 million environment steps, adjustments were made for the number of training instances based on the maximum steps per episode for different tasks: 1000 iterations for ant, halfcheetah, hopper, swimmer, and walker2d, while reacher was set to 20000 iterations.
> RLRD has high sample utilization but takes more time when converting eligible offline trajectory segments into current policy segments. The code repository for RLRD can be found at [2], using default parameters.
> SACAS is derived from the SAC algorithm, with the code available at [3], using default parameters. In cases where the environment encounters a fixed delay $d$, we modified the input dimensions of the SAC policy network and value function network to be the sum of the dimensions of the original state space |S| and the original action space |A| scaled by the delay $d$, denoted as |S| + d|A|. When the environment encounters a random delay $z_t$, we modified the input dimensions of the SAC policy network and value function network to be the sum of the dimensions of the original state space |S| and the maximum delay |$d_I+d_M$| times the action dimensions |A|, expressed as |S| + |$d_I+d_M$||A|.
>
> >**[Q5]** In the end of the first paragraph, it was mentioned minor delays markedly amplifies the risk in self-driving. Is there any reference? Following this question, are there more examples (better with data) of practical scenarios that suffers heavily from delays?
>
> **[A5]** In the '5G + Intelligent Transportation' series white paper (link provided below [4]), clear indicators are specified for the latency requirements in four scenarios of intelligent driving, as detailed below:
> Scenario	Assisted Driving	Remote Driving	Autonomous Driving	Vehicle Convoy
>
> | Scenario             | Assisted Driving                             | Remote Driving | Autonomous Driving | Vehicle Convoy   |
> |----------------------|----------------------------------------------|----------------|--------------------|------------------|
> | Maximum Latency (ms) | 20ms/100ms（V2V） 100ms（V2I） 1000ms（V2N） | 20ms           | 3ms-100ms          | 10ms-25ms        |
> |                      |                                              |                |                    |                  |
>
> Additionally, in literature [5], practical observations indicate a delay of 0.2 to 0.5 seconds between steering signals from the vehicle's computer and the actual steering wheel operation. While the vehicle remains in motion, this leads to substantial deviations in path tracking, potentially resulting in a loss of control when traveling at high speeds. Furthermore, literature [6] notes that hydraulic vehicle braking systems typically require over 0.4 seconds to achieve the necessary deceleration, significantly impacting the planning and control modules of Connected and Autonomous Vehicles (CAVs).
> In various other domains such as control of robotic systems [7], distributed computing [8], and management of traffic networks [9], delays have demonstrated adverse effects on the performance of corresponding intelligent agents."
>
> **Reference**
>
> [1] https://github.com/baimingc/dambrl
>
> [2] https://github.com/rmst/rlrd
>
> [3] https://github.com/pranz24/pytorch-soft-actor-critic
>
> [4]http://13115299.s21d-13.faiusrd.com/0/1/ABUIABA9GAAglcTHjQYojIWZswY.pdf?f=%E4%B8%AD%E5%9B%BD%E8%81%94%E9%80%9A5G+%E8%BD%A6%E8%BD%BD%E5%AE%9A%E5%88%B6%E5%8C%96%E6%A8%A1%E7%BB%84+%E7%99%BD%E7%9A%AE%E4%B9%A62019.pdf&v=1639047701
>
> [5]赵建辉, 高洪波, 张新钰, 等. 基于时间延迟动态预测的自动驾驶控制[J]. 2018.
>
> [6]Bayan, Fawzi P., et al. "Brake timing measurements for a tractor-semitrailer under emergency braking." SAE International Journal of Commercial Vehicles 2.2009-01-2918 (2009): 245-255.
>
> [7]Imaida, Takashi, et al. "Ground-space bilateral teleoperation of ETS-VII robot arm by direct bilateral coupling under 7-s time delay condition." IEEE Transactions on Robotics and Automation 20.3 (2004): 499-511.
>
> [8]Hannah, Robert, and Wotao Yin. "On unbounded delays in asynchronous parallel fixed-point algorithms." Journal of Scientific Computing 76 (2018): 299-326.
>
> [9]Dollevoet, Twan, et al. "Delay propagation and delay management in transportation networks." Handbook of Optimization in the Railway Industry (2018): 285-317.

---

> ### Author Response · Authors · 2023-11-20
> **Response to  Reviewer HvM3 (Part 3/3)**
>
> >**[Q6]** Why the performance of DEER is sensitive to dimension of latent representation? In particular, why dim=512 results in significantly worse performance in Walker2D (Figure 6)? My expectation is large dimension should work better or similar.
>
> **[A6]** The context representations are obtained by encoding the information states through a pre-trained encoder, capable of depicting delayed state information, thus their quality significantly impacts the final performance of the algorithm. The dimensionality reflects the representational capacity of the context representation, making DEER sensitive to dimensions. In the literature mentioned by the reviewer, OFENet explicitly highlights that simply increasing the dimensionality of the decision space doesn’t necessarily lead to improved final performance, as performance is also influenced by network architecture design, regularization, and auxiliary task design. DEER conducts pre-training before policy training, and the encoder structure remains unchanged throughout the subsequent process, differing from OFENet's update framework. Additionally, in the walker2d task, a decrease in performance is observed with 512 dimensions, likely due to overfitting during the encoding and decoding in the pre-training phase.
>
> >**[Q7]** Minor: including the size of font in Figure 3, the color of linestyle in Figure 6 and Figure 7, a table summarizing the numerical reults and A typo in the title "resillient".
>
> **[A7]** The aforementioned content will be updated in my newly uploaded PDF.

---

> ### Author Response · Authors · 2023-11-22
>
> Has our response sufficiently addressed your inquiries? Are there any additional problems requiring discussion? We eagerly anticipate further correspondence with you. Please let us know if there are any additional comments, questions, or concerns. Many thanks again for the invaluable hard work of all reviewers to improve our work.

---

> ### Author Response · Authors · 2023-11-23
>
> In **[A1]**, we revisited and elaborated on our innovations. Within **[A2]**, under the random delay setup suggested by the reviewer, we conducted diverse algorithmic comparisons on Hopper and Walker2d. **[A3]** delineates the comparison between DEER and vanilla SAC. **[A4]** explicates the application specifics of the Baseline method. **[A5]** illustrates the hazards of delays in real-world scenarios. **[A6]** analyzes the reasons behind DEER's sensitivity to dimensions. We've also updated the graphs and charts in the newly uploaded PDF. We sincerely look forward to engaging in further discussions with you.

---

> ### Author Response · Authors · 2023-11-23
>
> We hope our response has satisfactorily addressed your queries and concerns. Please do not hesitate to reach out if you have any further questions or require additional clarification. We are truly grateful for the invaluable contributions made by the reviewer, which have significantly enhanced the quality of our work.

---

> > ### Comment · Reviewer_HvM3 · 2023-12-01
> > **Thanks for the reply**
> >
> > I appreciate the author's repsonses. However, my concerns have not been fully addressed. E.g., there is no reply to my last comment in Weakness, and my inquiry to evaluate DEER's performance in delay=0 has not been addressed by the authors (for my comment "DEER should also be compared with vanilla SAC in delay=0 case"). Also, it is still not explained why DEER outperforms expert very much in the Swimmer task.
> >
> > Overall, I decide to maintain my original rating and I believe a comprehensive revision of the paper will be helpful to improve this work.

---

### Official Review · Reviewer_b2a3 · 2023-11-02

**Soundness:** 3 good
**Presentation:** 4 excellent
**Contribution:** 2 fair
**Rating:** 5
**Confidence:** 4

**Summary:**

This paper targets a kind of decision-making task with delays. To this end, the authors propose Delay-resilient Encoder-Enhanced (DEER) RL, in which a Seq2seq model taking the delayed state and subsequent action sequence as inputs and the corresponding state sequence as outputs is trained using the offline dataset. Then, this model provides hidden vector encoding information without delay during the online stage. DEER can handle random delay length and has fixed a hidden dimension. DEER is tested on Mujoco benchmark and shows superior performance.

**Strengths:**

Tasks with delay often exist in practical scenarios, so it is valuable to propose effective methods for them. Overall, the general idea behind DEER is simple and clear. Leveraging a pretrained model from offline dataset to provide information without delay during the online stage is reasonable. DEER shows stronger performance than compared baselines. The presentation in this paper is pretty good.

**Weaknesses:**

Experiments can be improved in the following aspects.
- Currently no baselines use the offline dataset, but DEER uses offline data to train the seq2seq model. Some offline-to-online methods should be included as baselines.
- Offline dataset’s influence on the model performance is not discussed and analyzed. Offline dataset plays a critical role in DEER, so it should be analyzed.

Some shortcomings of previous methods, like inference time, model precision and cumulative errors also exist in DEER. DEER uses an RNN which is not very efficient in terms of the inference time. RNN also has cumulative errors. Except for using more data (the offline dataset), why is DEER more effective than previous methods?

**Questions:**

Why does pretrained encoder have a very specific requirement (generated by a random policy along with a few expert trajectories) on the offline trajectories?

---

> ### Author Response · Authors · 2023-11-20
> **Response to Reviewer b2a3 (Part 1/2)**
>
> We are glad that the reviewer characterized our work as “clear” and “reasonable”. We thank the reviewer for the useful and detailed comments. In the following, we aim to address your questions and concerns. For the convenience of the reviewer, we reiterate the comments below.
> >**[Q1]** Currently no baselines use the offline dataset, but DEER uses offline data to train the seq2seq model. Some offline-to-online methods should be included as baselines.
>
>  **[A1]**  In response to the reviewer's inquiry, we undertook a comparison between the state-of-the-art algorithms PEX [1] and DEER. PEX's core approach involves initially learning a policy from an offline dataset, utilizing this learned policy as a candidate. Subsequently, another policy takes charge of further learning. Both policies interact with the environment in an adaptable manner. While training the offline policy, PEX requires reward information to guide its learning, in addition to states and actions. In contrast, DEER only relies on states and actions.
>
> Within the MuJoCo environment, we conducted policy training using PEX across six continuous tasks (Ant, HalfCheetah, Hopper, Reacher, Swimmer, and Walker2d), incorporating delays of 1, 2, 4, 6, and 8. Each scenario underwent experimentation using three different seeds. The offline policy training dataset for PEX aligns with that of DEER. Offline policies for various tasks were trained over a day. ~~Presently, the online training has persisted for two and a half days, set to conclude tonight. The comparative results between PEX and DEER will be updated here upon completion of the training.~~ The training outcomes for the Ant, Hopper,  Walker2d, and **HalfCheetah** tasks have been updated. The comparison between the training results of PEX and DEER for these three tasks is shown in the table below.
>
> |    Delay    |   1   |      |   2  |      |   4  |       |   6  |      |   8  |      |
> |:-----------:|:-----:|:----:|:----:|:----:|:----:|:-----:|:----:|:----:|:----:|:----:|
> |  Algorithm  |  DEER |  PEX | DEER |  PEX | DEER |  PEX  | DEER |  PEX | DEER |  PEX |
> |     Ant     |  4636 |  165 | 2903 |  111 | 2574 | 179.9 | 1653 |  314 | 1072 |  182 |
> |    Hopper   |  2484 |  8.6 | 3096 |  6.9 | 2918 |  7.7  | 2565 |  6.5 | 2462 |  7.1 |
> |   Walker2d  |  5411 | -5.1 | 4783 | -4.8 | 4119 |  1.6  | 3546 | -4.8 | 3074 | -4.5 |
> | HalfCheetah | 11971 | 1315 | 9352 | 1058 | 5780 |  1038 | 3853 |  957 | 2924 |  860 |
>
> Analysis of the data in the table indicates that the final training outcome of PEX closely resembles that of an agent employing a random strategy. The subpar performance is likely associated with the utilized offline dataset, primarily constituted by a significant majority of random trajectories alongside a limited number of expert trajectories. This composition within the offline data has resulted in PEX learning a strategy leaning towards randomness.
>
>
> >**[Q2]** Why does pretrained encoder have a very specific requirement (generated by a random policy along with a few expert trajectories) on the offline trajectories?  Offline dataset’s influence on the model performance is not discussed and analyzed. Offline dataset plays a critical role in DEER, so it should be analyzed.
>
> **[A2]** The reason behind structuring the offline dataset for pre-training the encoder to include a substantial amount of random trajectories and a few expert trajectories in the delay-free environment is to reflect a more realistic scenario, where random trajectories are predominant while expert trajectories are comparatively scarce. To address the reviewer's inquiry regarding the impact of the offline dataset on the model, we introduced two additional types of offline datasets in the Hopper and Walker2d tasks, both with a delay of 4: one composed entirely of random trajectories and the other exclusively of expert trajectories. The subsequent policy training procedures align with DEER's methodology. The comparative outcomes from training with these three distinct offline datasets are presented in the table below.
>
> | Offline datasets (different kinds of trajectories) | Random         | Large random and few expert | Expert         |
> |----------------------------------------------------|----------------|-----------------------------|----------------|
> | Hopper                                             | 2943.56±891.35 | 2988.32±432.15              | 3068.09±617.05 |
> | Walker2d                                           | 405.98±110.6   | 2289.55±968.70              | 510.67±95.25   |
> |                                                    |                |                             |                |
>
> The experimental results from the Walker2d task in the table reveal an intriguing aspect: the quantity of expert trajectories doesn't directly correlate with superior final results. Essentially, the efficacy of DEER's pre-trained encoder seems to hinge closely on the distribution of state transitions within the provided dataset.

---

> > ### Author Response · Authors · 2023-11-21
> > **Response to Reviewer b2a3 (Part 2/2)**
> >
> > >**[Q3]** Some shortcomings of previous methods, like inference time, model precision and cumulative errors also exist in DEER. DEER uses an RNN which is not very efficient in terms of the inference time. RNN also has cumulative errors. Except for using more data (the offline dataset), why is DEER more effective than previous methods?
> >
> > **[A3]** DEER leverages existing data from delay-free environments to learn an encoder-decoder for multi-step prediction. During the decision-making phase, DEER employs the encoder's representation of input states and action sequences, termed as context representation, rather than the final state from multi-step predictions. This representation extracts decision-relevant information from the most recent observed states and action sequences, thereby mitigating sensitivity to deviations from the true state caused by model precison and cumulative errors.
> >
> > **Reference**
> > [1] Zhang, Haichao, Wei Xu, and Haonan Yu. "Policy Expansion for Bridging Offline-to-Online Reinforcement Learning." The Eleventh International Conference on Learning Representations. 2022.

---

> > ### Author Response · Authors · 2023-11-23
> > **Updata for comparison of PEX and DEER**
> >
> > The latest updated data is presented in the table below. Experiments are still ongoing for the Swimmer and Reacher tasks. Once these experiments are completed, the data will also be updated.
> >
> > |    Delay    |   1   |      |   2  |      |   4  |       |   6  |      |   8  |      |
> > |:-----------:|:-----:|:----:|:----:|:----:|:----:|:-----:|:----:|:----:|:----:|:----:|
> > |  Algorithm  |  DEER |  PEX | DEER |  PEX | DEER |  PEX  | DEER |  PEX | DEER |  PEX |
> > |     Ant     |  4636 |  165 | 2903 |  111 | 2574 | 179.9 | 1653 |  314 | 1072 |  182 |
> > |    Hopper   |  2484 |  8.6 | 3096 |  6.9 | 2918 |  7.7  | 2565 |  6.5 | 2462 |  7.1 |
> > |   Walker2d  |  5411 | -5.1 | 4783 | -4.8 | 4119 |  1.6  | 3546 | -4.8 | 3074 | -4.5 |
> > | HalfCheetah | 11971 | 1315 | 9352 | 1058 | 5780 |  1038 | 3853 |  957 | 2924 |  860 |

---

> ### Author Response · Authors · 2023-11-22
>
> Has our response sufficiently addressed your inquiries?
> Are there any additional problems requiring discussion?
> We eagerly anticipate further correspondence with you.
> Please let us know if there are any additional comments, questions, or concerns.
> Many thanks again for the invaluable hard work of all reviewers to improve our work.

---

> ### Author Response · Authors · 2023-11-23
>
> We hope our response has satisfactorily addressed your queries and concerns. Please do not hesitate to reach out if you have any further questions or require additional clarification. We are truly grateful for the invaluable contributions made by the reviewer, which have significantly enhanced the quality of our work.

---

### Author Response · Authors · 2023-11-21
**Summary of response**

We sincerely thank all the reviewers for their valuable and detailed feedback. We are glad that the reviewers found: (1) Our research problem is practical and important (all reviewers). (2) Our proposed method DEER is a reasonable and versatile approach which performs well in delayed tasks (all reviewers).

Based on the reviewers' comments we (A) revised the paper and clarified some unclear and possibly confusing expressions, especially our novelty and contributions, (B) conducted additional experiments to cover more settings and performance analysis. Here, we list the main changes:

**Detailed analysis of our approach**
1. we clarified our contributions and the differences among them (Reviewer yS4B, HvM3) and the novelty of using Seq2Seq for delayed settings (Reviewer pa1n, b2a3).
2. we clarified more implementation details, such as Seq2Seq to handle delays (Reviewer yS4B, pa1n) and baseline models (Reviewer HvM3)

**More experiments**
1. We compared our method with the online variant (Reviewer yS4B) and an offline-to-online method (Reviewer b2a3) for fairness. We also conducted experiments to demonstrate the impact of dataset composition (Reviewer b2a3).
2. We evaluated our method on a new random delayed setting inspired by Reviewer HvM3, and the result shows that our method can perform well on both original and new settings.
3. We made more experiments on our key hyper-parameter, the embedding dimension (Reviewer HvM3, pa1n) .

Please let us know if there are any additional comments, questions, or concerns. Many thanks again for the invaluable hard work of all reviewers to improve our work.

---

### Meta-Review · Area_Chair_WEta · 2023-12-10

**Metareview:**

The paper proposes Delay-resilient Encoder-Enhanced RL (DEER) framework to handle delayed observations via an encoder to map observations and actions into a latent state, Various experiments on MuJoCo robotic movement tasks are used to evaluate DEER.

Most of the reviewers are unconvinced about the paper, and one reviewer raising his rating to marginally above acceptance threshold after the rebuttal. The biggest criticism are lack of major technical innovation and lack of comparison with existing baselines.

During the rebuttal phase the authors did some more experiments - however, there were still important experiments mission (e.g. SAC with delay zero, as specifically called out by one reviewer). Upon highlighting this mission experiment authors finally provided some numbers - however the numbers are inconclusive. I agree with reviewer HvM3 that the paper needs to be revised and submitted to another venue.

**Justification For Why Not Higher Score:**

A majority of the reviewers recommend rejecting the paper.

**Justification For Why Not Lower Score:**

N/A

---

### Decision · Program_Chairs · 2024-01-16

Reject